# Objective differences and pathway differentiation between twin subtypes within the same IDDSI level: A texture–nutrition analysis based on commonly consumed foods among Chinese elderly

Muxi Chen●[1,2], Yi Cheng[1], Juan Duan[1], Lei Shi[1], Yuan Liu[1]*

**1** Department of Clinical Nutrition, West China Hospital, Sichuan University, Chengdu, China, **2** Clinical Medicine College, College of Traditional Chinese Medicine, Mianyang, China

* liu63415@163.com

## Abstract

### Objective

This study adopts a "pathway-priority" perspective to systematically quantify the texture–moisture–nutrition continuum of elderly-preferred food ingredients commonly consumed in China under the *International Dysphagia Diet Standardisation Initiative* (IDDSI) framework. It focuses on intra-level differentiation between "twin-type/pathway pairs" (MO3 vs. LQ3, EX4 vs. PU4, EC7 vs. RG7), aiming to reveal objective distinctions in texture and nutritional orientation across pathways and to provide quantitative evidence for dietary level classification and prescription substitution in dysphagia management.

### Methods

Twenty-six representative food ingredients were collected from seven major Chinese dietary regions through Python-based web crawling. According to IDDSI classification, samples were grouped into the *liquid pathway* (Levels 0–2, MO3, EX4) and *food pathway* (LQ3, PU4, Levels 5–7 including EC7/RG7). A TA.New Plus texture analyzer was used to measure hardness, adhesiveness, and cohesiveness. Nutritional components per 100 g—including energy, protein, fat, carbohydrate, and dietary fiber—were calculated using *Huaxi Hospital Nutrition* software. Statistical analyses included Mann–Whitney U tests with Cliff's δ for twin-type comparisons, Spearman correlation and Theil–Sen slope analysis for monotonic trends within pathways, and Kruskal–Wallis followed by Dunn–Bonferroni tests for intra-level differences among food categories. False discovery rate (FDR–BH) correction was applied, with significance set at $q \leq 0.05$.

**Data availability statement:** All relevant data are within the paper and its Supporting information files.

**Funding:** Funding:Y.L. received funding from the project "Key Technologies and New Product Development for Intelligent and Safe Production of Foods for Special Medical Purposes" (Grant No. JH2019045).L.S. received funding from the project "Development and Industrialization Demonstration of Nutrition and Health Foods for the Elderly" (Grant No. 2023YFF1104405), funded by the Ministry of Science and Technology of China.

**Competing interests:** The authors have declared that no competing interests exist.

## Results

(1) In twin-type comparisons, MO3, EX4, and EC7 exhibited significantly lower hardness and energy density but higher moisture content (q ≤ 0.05), whereas LQ3, PU4, and RG7 contained higher energy and protein levels, reflecting hydration-safety orientation in liquid pathways versus structural-nutritional orientation in food pathways. (2) Within-pathway analyses showed that the liquid pathway (Levels 0→4) followed a "thickening–hardening–dehydrating" monotonic trend, with increasing hardness and energy but decreasing adhesiveness and moisture. The food pathway (Levels 3→7) displayed a "structuring–hardening–dehydrating–energizing" trajectory, indicating enhanced granularity and nutritional concentration. (3) Significant texture differences among food categories within the same level were observed ($\varepsilon^2 = 0.37$–$0.67$, $q ≤ 0.05$), suggesting that "same level ≠ interchangeable.". (4) The established "safety–nutrition balance window" identified key control zones: moisture 88.9–96.8 g/100 g and hardness 100–160 N/m² for MO3/EX4, and energy 70–324 kcal/100 g for PU4/RG7, providing practical references for clinical and process optimization.

## Conclusion

Twin subtypes within the same IDDSI level exhibit systematic differences in texture and nutrition, confirming the hypothesis that "same level ≠ interchangeable." The liquid pathway emphasizes safety and hydration, while the food pathway prioritizes structure and nutrient density. By constructing an intra-pathway "texture–nutrition continuum" and safety–nutrition balance window, this study offers quantitative evidence for pathway selection, diet prescription, and quality control in dysphagia management. Future work should integrate VFSS/FEES imaging and clinical outcomes to map laboratory metrics to swallowing safety, facilitating translation from experimental assessment to clinical application.

## 1. Introduction

China has rapidly entered an aging society, accompanied by growing health demands among older adults. Swallowing function gradually declines during aging and under chronic conditions such as stroke, Parkinson's disease, Alzheimer's disease, and post–head and neck cancer therapy, leading to varying degrees of dysphagia. The prevalence of dysphagia increases markedly with age and is particularly high among specific patient populations. Studies have reported a prevalence of 66.0% among elderly Chinese, including 21.0% at ages 60–69, 28.0% at ages 70–79, and 41.0% at ≥80 years [1]. Dysphagia occurs in 50–60% of patients following head and neck cancer treatment and in 37–78% of post-stroke patients [2,3]. Dysphagia alters dietary structure and eating patterns, impairs hydration and nutrient intake, and is closely associated with aspiration pneumonia, malnutrition, frailty, and reduced quality of life [4]. Within the continuum of "healthy aging" and integrated home–institution–medical

care, developing diets that balance safety, nutrition, and compliance across functional states has become an urgent clinical and public health challenge.

To standardize terminology and practice, the *International Dysphagia Diet Standardisation Initiative* (IDDSI) established an eight-level continuum (Levels 0–7) for both drinks and foods, with standardized testing for texture and consistency [5]. The drink pathway primarily covers Levels 0–4 (0: thin; 1: slightly thick; 2: mildly thick; 3: moderately thick; 4: extremely thick), while the food pathway encompasses Levels 3–7 (3: liquidized; 4: pureed; 5: minced & moist; 6: soft & bite-sized; 7: regular/easy to chew). Natural overlaps occur at Levels 3 and 4, where the same numeric level can correspond to either a drink or a food target. At Level 7, the "regular" (RG7) type is complemented by an "easy to chew" (EC7) subtype, representing a lower bite force threshold and better oral breakdown properties [6].

Importantly, a numeric level does not imply product equivalence.First, within Levels 3 and 4, two distinct testing and swallowing-kinematic pathways coexist—here termed "twin subtypes" or "dual pathways." Level 3 includes MO3 (drink pathway, moderately thick) and LQ3 (food pathway, liquidized), while Level 4 includes EX4 (drink pathway, extremely thick) and PU4 (food pathway, pureed). Level 7 includes RG7 and EC7, both food types, the latter optimized for limited mastication.Substituting products solely by numeric level neglects key differences between syringe flow tests (drinks) and spoon/fork pressure tests (foods), resulting in mismatches in aspiration risk, hydration adequacy, and energy density [7]. Second, empirical and clinical evidence indicates marked heterogeneity in hardness, adhesiveness, and cohesiveness among food ingredients within the same level, owing to variations in food matrix, fiber type and content, starch gelatinization/retrogradation, protein gelation, and fat emulsification [8–12]. Thus, "same level" does not guarantee interchangeability. Pathway-based intra-level classification may better explain such differences and guide precise clinical selection.

In the Chinese context, elderly-preferred food ingredients differ substantially in matrix composition, colloidal behavior, fiber structure, and water-holding capacity. Consequently, foods at the same IDDSI level may exhibit distinct "texture–nutrition" profiles depending on ingredient type and processing. Providing quantitative, instrument-based evidence to support IDDSI bedside testing not only addresses the uncertainty of intra-level substitution but also supports a "pathway-first, within-pathway substitution" approach for diet prescription and quality control [10,14].

Accordingly, this study proposes a *pathway-priority* framework for IDDSI sub-classification: explicitly distinguishing the drink and food pathways within the same numeric level, focusing on twin subtypes at Levels 3, 4, and 7 (L3: MO3 vs. LQ3; L4: EX4 vs. PU4; L7: EC7 vs. RG7), and mapping the full 0–7 continuum within each pathway. Using commonly consumed and elderly-preferred Chinese food ingredients, we systematically measured and compared hardness, adhesiveness (signed), cohesiveness, moisture/addition ratio, and macronutrient composition (energy, protein, fat, carbohydrate, dietary fiber), stratified by food category.

The study aimed to:

(1) quantify objective differences and pathway differentiation between twin subtypes within the same IDDSI level;

(2) construct intra-pathway "texture–nutrition" continua; and

(3) provide data-driven, exploratory evidence for practical issues such as initial feeding form (liquid-first vs. solid-first) and rehabilitation sequencing, thereby defining operable substitution boundaries for clinical and catering applications in China.

## 2. Materials and methods

### 2.1 Ingredient data collection

To better understand dietary preferences among the Chinese elderly, this study used the Python programming language and its web-scraping library Scrapy to collect data from Meishi Tianxia (www.meishichina.com), a widely used Chinese culinary website. Data were extracted from elderly-recommended recipes covering seven major Chinese dietary regions.

   

In compliance with the website's access policy and relevant legal regulations, information on major food ingredients from each recipe was retrieved. The dataset was then cleaned and filtered to remove duplicates and irrelevant content, retaining only key variables (ingredient name, category, quantity, and cooking method) for subsequent analysis.To clarify the relationship between web-derived information and laboratory experimentation, web crawling was used exclusively to identify commonly consumed food ingredients, elderly-friendly preparation practices, and regional dietary representation. All texture measurements reported in this study were conducted on physically prepared samples rather than web-derived data. Selected food ingredients were freshly purchased from local markets and processed in the laboratory using standardized steaming/cooking and blending procedures described below. Thus, web-scraped information served only for ingredient selection and contextual dietary relevance, whereas texture and nutritional analyses were based entirely on experimentally prepared samples.

## 2.2 Selection of experimental food ingredients

Using web-scraped data, 26 representative food ingredients commonly consumed and recommended for older adults across China's seven dietary regions were selected.

These food ingredients included:Meats: pork, beef, chicken, fish;Eggs: chicken eggs;Grains and tubers: noodles, rice, oats, potatoes, sweet potatoes, yam, pumpkin;Vegetables: tomato, eggplant, Chinese cabbage, carrot, green cabbage, broccoli, spinach;Mixed beans: red beans, mung beans;Fruits: apple, pear, peach, banana;Other: tofu

These items are culturally representative, widely accessible, and considered elderly-friendly in the Chinese dietary context.The web-derived information was used primarily for ingredient screening and contextual dietary reference rather than as a source of experimental measurements.

## 2.3 Food processing and sample preparation

All food ingredients were processed following typical cooking methods used in elderly diets to ensure standardization. General preparation involved cleaning, cutting, and steaming until fully cooked, followed by blending with varying amounts of water to produce purees with different consistencies and hardness levels.

For example, "pumpkin puree (1:2)" indicates a mass ratio of 1 part pumpkin to 2 parts water.Specific procedures were as follows:

Meat samples: Fresh pork, beef, and fish were trimmed of fascia and fat, cut into 2 × 2 cm cubes, minced, steamed for 30 min, and blended with different water ratios to form purees (e.g., pork puree (1:0) means no added water).

Egg samples: Hard-boiled egg yolks were mixed with varying water ratios (e.g., yolk (1:0.5) = yolk:water = 1:0.5). For steamed eggs, beaten eggs were mixed with water and steamed (e.g., steamed egg (1:2) = egg:water = 1:2).

Grains and tubers: Rice was soaked for 30 min and steamed with a 1:4 rice–water ratio; oats were cooked at 1:8; noodles were boiled for 8 min, minced, and mixed with water (1:1).

Vegetables: Cleaned and diced (2 × 2 cm) vegetables were steamed for 30 min and blended (e.g., carrot puree (1:0.5) = carrot:water = 1:0.5).

Mixed beans: Red and mung beans were soaked for 12 h, steamed for 30 min, and blended (e.g., red bean puree (1:2) = bean:water = 1:2).

Fruits: Peeled, cored, diced (2 × 2 cm) fruits were steamed or used raw and blended with water (e.g., apple puree (1:1) = apple:water = 1:1).

Other: Tofu was steamed for 15 min, then blended directly or with small amounts of water depending on the desired consistency.

All samples followed a uniform preparation protocol to ensure comparability. Three independent replicates (50 g each) were prepared per ingredient at each consistency level for IDDSI grading and texture analysis.

## 2.4 IDDSI classification and pathway definition

**(1) IDDSI grading procedures and functional criteria.** All prepared food-ingredient samples were classified according to the International Dysphagia Diet Standardisation Initiative (IDDSI) framework using standardized bedside assessment methods. These included the IDDSI Flow Test (10-mL syringe, 10-s gravity flow), Fork Drip Test, Spoon Tilt Test, and Fork Pressure Test, depending on whether the sample belonged to drink or food categories.

For drinks (Levels 0–4), classification followed the remaining volume criteria after 10 s: Level 0 (Thin) = 0–1 mL remaining; Level 1 (Slightly thick) = 1–4 mL; Level 2 (Mildly thick) = 4–8 mL; Level 3 (Moderately thick drink) = 8–10 mL; Level 4 (Extremely thick drink) = 10 mL remaining (no flow).

For foods (Levels 3–7), classification relied on spoon/fork functional performance: Level 3 liquidized foods required cohesive flow without separation; Level 4 pureed foods held shape on a spoon and slid off with minimal residue; Levels 5–7 foods were assessed using fork pressure and chewing requirements. Each prepared sample was required to pass the appropriate IDDSI functional test(s) before inclusion in subsequent texture and nutritional analyses.

**(2) Definition of twin subtypes and mapping to official IDDSI terminology.** To clarify potential ambiguity between IDDSI food and drink categories sharing the same numeric level, this study defined "twin subtypes" as paired samples with identical IDDSI level numbers but belonging to different IDDSI functional categories. These subtype labels correspond to official IDDSI terminology as follows:

MO3 – Moderately thick drink (IDDSI Drinks Level 3; liquid pathway), verified by Flow Test (8–10 mL remaining).

LQ3 – Liquidized food (IDDSI Foods Level 3; food pathway), verified by spoon/fork criteria.

EX4 – Extremely thick drink (IDDSI Drinks Level 4; liquid pathway), verified by Flow Test (10 mL remaining).

PU4 – Pureed food (IDDSI Foods Level 4; food pathway), verified by Spoon Tilt/Fork Drip criteria.

EC7 – Easy-to-chew food (IDDSI Level 7 easy-to-chew subtype).

RG7 – Regular food (IDDSI Level 7 regular texture).

This classification does not modify IDDSI definitions; it serves only as an analytical framework to describe within-level variability.

**(3) Analytical pathway integration.** For trend analysis, samples were grouped into two functional pathways reflecting IDDSI testing logic:

Liquid pathway: Levels 0–2, MO3, and EX4 (primarily flow-test based).

Food pathway: LQ3, PU4, Levels 5–7 (spoon/fork pressure-based).

This pathway grouping was used solely for comparative analysis of texture–nutrition relationships and does not imply reclassification of IDDSI levels.

## 2.5 Texture measurements

Texture characteristics were measured using a TA.New Plus texture analyzer (ISENSO, USA). Parameters included hardness, cohesiveness, and adhesiveness, obtained via Texture Profile Analysis (TPA) in double-compression mode.

Testing conditions were standardized as follows:

Mode: Compression (distance-controlled)

 

Target distance: 3.000 mm

Pre-test and return speed: 0.50 mm/s

Trigger force: 3.000 gf

Prior to testing, samples were gently de-aerated after blending and allowed to equilibrate at room temperature (25 ± 1 °C) for approximately 10 min to minimize temperature- and air-related variability. Samples were transferred into identical containers with a consistent fill height to reduce geometric effects during compression testing. Whenever possible, samples were prepared and tested on the same day following a standardized operational protocol.

Each sample was tested three times, and mean values were used as final texture parameters. All measurements were conducted at 25 ± 1 °C to ensure comparability, reproducibility, and measurement stability.

### 2.6 Nutritional composition analysis

Nutritional composition per 100 g of each sample was estimated using Huaxi Hospital Nutrition software, a professional dietary analysis tool based on validated food composition databases. Calculations were performed using standardized ingredient weights and recorded water-addition ratios applied during sample preparation.

Because the experimental design involved single food ingredients processed by steaming/cooking and controlled blending with added water to achieve different IDDSI texture levels, nutrient values expressed per 100 g reflect both intrinsic ingredient composition and moisture-related dilution effects. This dilution effect represents a controlled experimental variable rather than a measurement artifact.

To improve interpretability across samples with varying moisture contents, additional density-related indicators (e.g., energy density expressed as kcal/g and relative protein density expressed as g protein per 100 kcal) were derived to support pathway-oriented nutritional interpretation without requiring chemical compositional assays.

### 2.7 Statistical analysis

(1) Descriptive statistics: Texture parameters (hardness, adhesiveness, cohesiveness), moisture content, and nutritional indices (energy, protein, etc.) were summarized by IDDSI level × subtype × food category and presented as median [IQR].

(2) Twin-type comparisons: Differences between twin subtypes (MO3 vs LQ3, EX4 vs PU4, EC7 vs RG7) were assessed using the Mann–Whitney U test with effect size (Cliff's δ) and Benjamini–Hochberg FDR correction ($q \leq 0.05$).

(3) Within-pathway monotonic trends: Level order was treated as an ordinal variable; Spearman's ρ and Kendall's τb were computed, and the Theil–Sen slope was used to estimate the average change per level. Pathway trajectories were visualized accordingly.

(4) Intra-level food-category differences: For key levels (L3, L4, L7), Kruskal–Wallis tests compared major food categories. When significant ($p < 0.05$), Dunn–Bonferroni pairwise comparisons followed. Effect size ($\varepsilon^2$) quantified group difference strength, and chi-square tests evaluated category distribution bias across subtypes.

(5) Safety–nutrition balance window: For key nodes (MO3, EX4, PU4, EC7, RG7), P25–P75 (target zone) and P10–P90 (tolerance zone) ranges were calculated for hardness, adhesiveness, moisture, energy, and protein. Confidence intervals (95%) were estimated via bootstrap resampling (n = 1000) to define balance windows.

(6) Software and significance: All analyses were performed in Python 3.9 (pandas, scipy, pingouin, matplotlib) and SPSS 26.0. Significance was determined by two-tailed $p < 0.05$, with $q \leq 0.05$ after multiple testing correction.

## 2.8 Data and ethical statement

This study involved no human or animal experimentation. All food materials were commercially sourced, and the experimental preparation process complied with the Laboratory Food Safety Code of Practice.

## 3. Results

### 3.1 Sample characteristics and composition

A total of 585 food samples were included in this study, comprising 301 samples (51.5%) classified under the liquid pathway and 284 samples (48.5%) under the food pathway.According to the International Dysphagia Diet Standardisation Initiative (IDDSI) framework, the liquid pathway covered Levels 0–2 and the subtypes MO3 and EX4, while the food pathway included subtypes LQ3, PU4, and Levels 5–7 (including EC7 and RG7).The sample distribution across IDDSI levels and subtypes was as follows:Level 0–110 samples (18.8%), Level 1–84 samples (14.4%), Level 2–78 samples (13.3%).Within Level 3, MO3 accounted for 24 samples (4.1%) and LQ3 for 57 samples (9.7%).Within Level 4, EX4 included 5 samples (0.9%) and PU4 included 50 samples (8.5%).Higher levels consisted of Level 5–63 samples (10.8%), Level 6–78 samples (13.3%), Level 7 EC7–21 samples (3.6%), and RG7–15 samples (2.6%).

The major food-category compositions within each subtype were as follows:Level 0: mainly fruit (32.7%) and vegetable (26.4%).Level 1: dominated by vegetable (39.3%) and fruit (28.6%).Level 2: composed primarily of vegetable (26.9%), fruit (19.2%), and egg (19.2%).Level 3: MO3 was dominated by vegetable (58.3%), while LQ3 was characterized by grains and tubers (35.1%) and vegetable (17.5%).Level 4: EX4 consisted mainly of vegetable (60.0%) and fruit (40.0%); PU4 contained vegetable (36.0%), grains and tubers (26.0%), and egg (24.0%) as primary components.

Level 5: mainly grains and tubers (33.3%), meat (19.0%), and vegetable (19.0%).

Level 6: vegetable (26.9%) and grains and tubers (26.9%) shared similar proportions.

Level 7: EC7 featured fruit (42.9%) and vegetable (28.6%), while RG7 showed a balanced distribution across mixed beans (20.0%), fruit (20.0%), and meat (20.0%).

A summary of the sample composition across IDDSI subtypes is presented in Table 1, and the proportional distribution of major food categories within each subtype is illustrated in Fig 1.

### 3.2 Comparison between twin subtypes/ dual pathways

This section compares paired subtypes within the same numeric IDDSI levels but belonging to different ingestion pathways, including Level 3 (MO3 vs. LQ3), Level 4 (EX4 vs. PU4), and Level 7 (EC7 vs. RG7). For each pair, texture parameters (hardness, adhesiveness, cohesiveness), hydration indices (water content and water ratio), and nutritional indicators (energy, protein, fat, carbohydrate, and dietary fiber per 100 g) were analyzed. Results are reported as median [IQR] along with $p$ values (Mann–Whitney U test), $q$ values (Benjamini–Hochberg correction), and effect sizes (Cliff's δ).

**Level 3 (MO3 vs. LQ3):** MO3 (n = 24) and LQ3 (n = 57) exhibited distinct textural and nutritional profiles. Hardness (118.60 vs. 143.41 N/m², $q > 0.05$) and cohesiveness (0.76 vs. 0.76, $q > 0.05$) were comparable, while adhesiveness tended to be lower in LQ3. Water content was significantly higher in MO3 (96.48 vs. 89.20 g/100 g, $q < 0.05$, δ = 0.622), indicating greater hydration. In contrast, LQ3 showed higher energy (51.00 vs. 11.20 kcal/100 g, $q < 0.05$), protein, fat, and carbohydrate contents, suggesting denser nutritional composition in the food pathway.

**Level 4 (EX4 vs. PU4):** EX4 (n = 5) and PU4 (n = 50) both represented thickened consistencies but differed in pathway classification. PU4 exhibited higher hardness (160.96 vs. 130.50 N/m², $q > 0.05$) and nutrient density, while EX4 showed significantly greater water content (96.80 vs. 88.45 g/100 g, $q < 0.05$) and water ratio (1.50 vs. 0.50, $q < 0.05$).

**Table 1. Sample composition across IDDSI subtypes (N = 585).**

| IDDSI level/ subtype | n | Percent (%) | Path | Main food category composition (%) |
|---|---|---|---|---|
| 0 | 110 | 18.8 | Liquid-path | Fruit 32.7%; Vegetable 26.4%; Meat 13.6%; Other 12.7%; Grains and tubers 9.1%; Egg 4.5%; Mixed beans 0.9% |
| 1 | 84 | 14.4 | Liquid-path | Vegetable 39.3%; Fruit 28.6%; Grains and tubers 25.0%; Meat 3.6%; Other 3.6% |
| 2 | 78 | 13.3 | Liquid-path | Vegetable 26.9%; Fruit 19.2%; Egg 19.2%; Grains and tubers 17.9%; Meat 9.0%; Mixed beans 3.8%; Other 3.8% |
| MO3 | 24 | 4.1 | Liquid-path | Vegetable 58.3%; Grains and tubers 16.7%; Fruit 12.5%; Meat 4.2%; Egg 4.2%; Other 4.2% |
| LQ3 | 57 | 9.7 | Food-path | Grains and tubers 35.1%; Vegetable 17.5%; Fruit 15.8%; Meat 14.0%; Mixed beans 8.8%; Other 5.3%; Egg 3.5% |
| EX4 | 5 | 0.9 | Liquid-path | Vegetable 60.0%; Fruit 40.0% |
| PU4 | 50 | 8.5 | Food-path | Vegetable 36.0%; Grains and tubers 26.0%; Egg 24.0%; Meat 8.0%; Fruit 4.0%; Other 2.0% |
| 5 | 63 | 10.8 | Food-path | Grains and tubers 33.3%; Meat 19.0%; Vegetable 19.0%; Fruit 11.1%; Egg 7.9%; Mixed beans 6.3%; Other 3.2% |
| 6 | 78 | 13.3 | Food-path | Vegetable 26.9%; Grains and tubers 26.9%; Mixed beans 16.7%; Meat 14.1%; Fruit 9.0%; Egg 3.8%; Other 2.6% |
| EC7 | 21 | 3.6 | Food-path | Fruit 42.9%; Vegetable 28.6%; Grains and tubers 28.6% |
| RG7 | 15 | 2.6 | Food-path | Mixed beans 20.0%; Fruit 20.0%; Meat 20.0%; Egg 13.3%; Grains and tubers 13.3%; Vegetable 6.7%; Other 6.7% |

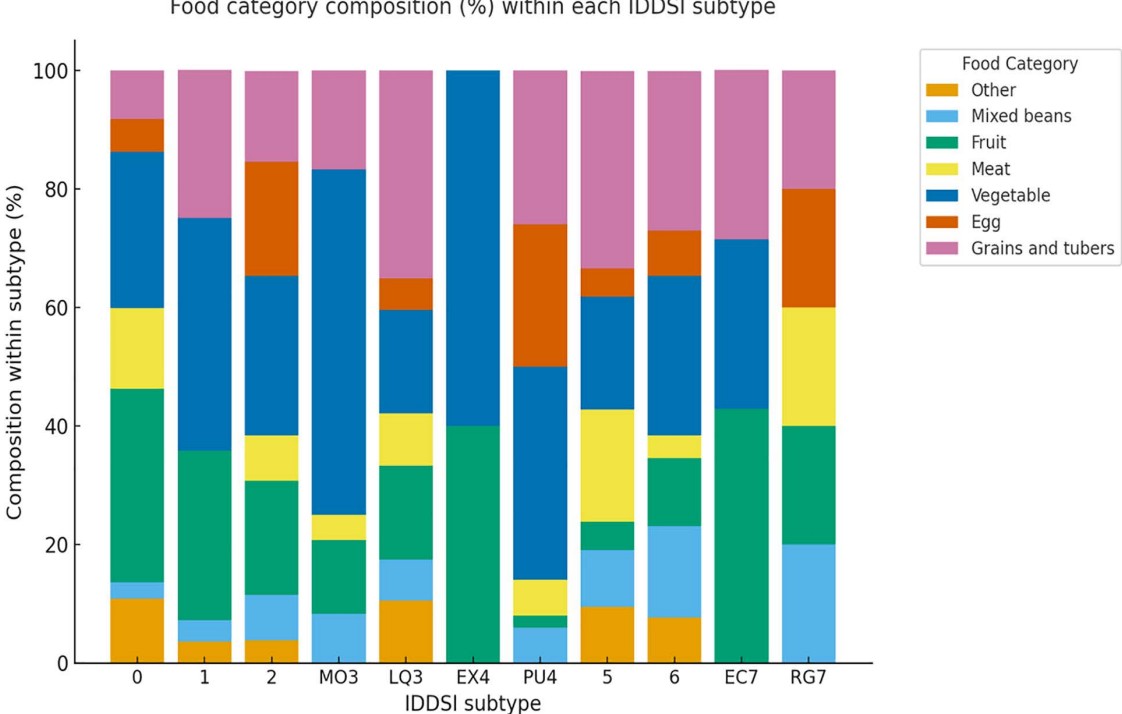

**Fig 1. Food category composition (%) within each IDDSI subtype.** The figure illustrates the proportional distribution of seven major food categories—grains and tubers, mixed beans, fruit, meat, vegetable, egg, and other (tofu)—within each IDDSI subtype, highlighting distinct ingredient patterns between the liquid and food pathways.

Nutritionally, PU4 contained higher energy (71.50 vs. 40.00 kcal/100 g, $q < 0.05$), protein, and fat levels, reflecting a trade-off between hydration safety and nutrient concentration.

**Level 7 (EC7 vs. RG7):** EC7 (n = 21) and RG7 (n = 15) both belong to the food pathway but differ in chewing thresholds. RG7 showed higher hardness (187.47 vs. 150.40 N/m², $q < 0.05$), while adhesiveness and cohesiveness did not differ significantly. Water content was greater in EC7 (89.50 vs. 83.50 g/100 g, $q < 0.05$), facilitating easier bolus formation. Nutritionally, RG7 demonstrated higher energy (71.50 vs. 51.00 kcal/100 g, $q < 0.05$), protein (15.20 vs. 0.90 g/100 g, $q < 0.05$), and fat contents, indicating a denser and more structured composition.

The summarized statistics for all indicators—including median [IQR], $p$, $q$, and Cliff's δ values—are presented in Table 2, providing a comprehensive overview of within-level differences.Comparisons of texture parameters are illustrated in Fig 2 (Level 3), Fig 3 (Level 4), and Fig 4 (Level 7), while corresponding nutritional profiles are depicted in Figs 5–7.

Together, these findings highlight a consistent pattern of lower hardness and energy density in liquid-pathway subtypes and higher nutrient concentration in food-pathway counterparts, supporting the concept that "same level ≠ interchangeable."

**Table 2. Within-level comparison of paired IDDSI subtypes.**

| Level | Indicator | Subtype A Median [IQR] | Subtype B Median [IQR] | *p*-value | q-value | Cliff's δ |
|---|---|---|---|---|---|---|
| Level 3 (MO3 vs LQ3) | Hardness (N/m²) | 118.60 [116.76–134.53] | 143.41 [103.40–202.37] | >0.05 | >0.05 | –0.183 |
| | Adhesiveness (gf) | –6.00 [–9.10—1.49] | –14.58 [–20.89–0.00] | >0.05 | >0.05 | 0.171 |
| | Cohesiveness (unitless) | 0.76 [0.71–0.77] | 0.76 [0.66–0.94] | >0.05 | >0.05 | –0.137 |
| | Water content (g/100 g) | 96.48 [88.90–96.80] | 89.20 [85.50–93.92] | <0.05 | <0.05 | 0.622 |
| | Water ratio | 1.00 [0.50–1.50] | 1.00 [0.50–1.50] | >0.05 | >0.05 | –0.032 |
| | Energy (kcal/100 g) | 11.20 [9.33–42.00] | 51.00 [22.80–71.50] | <0.05 | <0.05 | –0.630 |
| | Protein (g/100 g) | 0.93 [0.67–1.13] | 1.13 [0.73–4.45] | <0.05 | <0.05 | –0.472 |
| | Fat (g/100 g) | 0.15 [0.12–0.16] | 0.20 [0.10–0.40] | <0.05 | <0.05 | –0.360 |
| | Carbohydrates (g/100 g) | 2.00 [1.50–10.10] | 4.96 [2.27–13.10] | <0.05 | <0.05 | –0.328 |
| | Dietary fiber (g/100 g) | 0.04 [0.00–0.62] | 0.40 [0.00–1.00] | >0.05 | >0.05 | –0.214 |
| Level 4 (EX4 vs PU4) | Hardness (N/m²) | 130.50 [127.48–134.04] | 160.96 [131.55–214.22] | >0.05 | >0.05 | –0.400 |
| | Adhesiveness (gf) | –17.65 [–22.21—12.60] | –10.40 [–17.00–3.30] | >0.05 | >0.05 | –0.320 |
| | Cohesiveness (unitless) | 0.80 [0.74–0.86] | 0.80 [0.70–0.89] | >0.05 | >0.05 | –0.080 |
| | Water content (g/100 g) | 96.80 [95.00–97.30] | 88.45 [82.10–92.30] | <0.05 | <0.05 | 0.720 |
| | Water ratio | 1.50 [1.50–2.00] | 0.50 [0.25–1.00] | <0.05 | <0.05 | 0.680 |
| | Energy (kcal/100 g) | 40.00 [31.80–47.60] | 71.50 [47.30–108.00] | <0.05 | <0.05 | –0.680 |
| | Protein (g/100 g) | 1.20 [0.60–1.30] | 4.20 [1.30–6.20] | <0.05 | <0.05 | –0.720 |
| | Fat (g/100 g) | 0.20 [0.10–0.30] | 0.60 [0.20–2.30] | <0.05 | <0.05 | –0.640 |
| | Carbohydrates (g/100 g) | 9.90 [8.40–12.80] | 13.10 [5.10–19.30] | >0.05 | >0.05 | –0.440 |
| | Dietary fiber (g/100 g) | 1.00 [1.00–1.30] | 0.30 [0.00–1.20] | >0.05 | >0.05 | 0.280 |
| Level 7 (EC7 vs RG7) | Hardness (N/m²) | 150.40 [133.34–169.62] | 187.47 [171.22–212.94] | <0.05 | <0.05 | 0.454 |
| | Adhesiveness (gf) | –2.70 [–6.10–0.00] | –3.40 [–9.40–0.00] | >0.05 | >0.05 | –0.133 |
| | Cohesiveness (unitless) | 0.77 [0.72–0.84] | 0.80 [0.70–0.87] | >0.05 | >0.05 | 0.308 |
| | Water content (g/100 g) | 89.50 [85.20–93.20] | 83.50 [79.30–86.10] | <0.05 | <0.05 | 0.543 |
| | Water ratio | 0.50 [0.50–0.50] | 0.50 [0.25–0.50] | >0.05 | >0.05 | –0.200 |
| | Energy (kcal/100 g) | 51.00 [31.00–71.50] | 71.50 [51.00–91.00] | <0.05 | <0.05 | –0.543 |
| | Protein (g/100 g) | 0.90 [0.40–1.40] | 15.20 [1.40–20.20] | <0.05 | <0.05 | –0.800 |
| | Fat (g/100 g) | 0.20 [0.10–0.20] | 0.60 [0.20–2.30] | <0.05 | <0.05 | –0.543 |
| | Carbohydrates (g/100 g) | 13.10 [9.90–26.20] | 13.10 [3.40–22.00] | >0.05 | >0.05 | 0.114 |
| | Dietary fiber (g/100 g) | 1.10 [1.00–2.60] | 0.10 [0.00–1.20] | >0.05 | >0.05 | 0.257 |

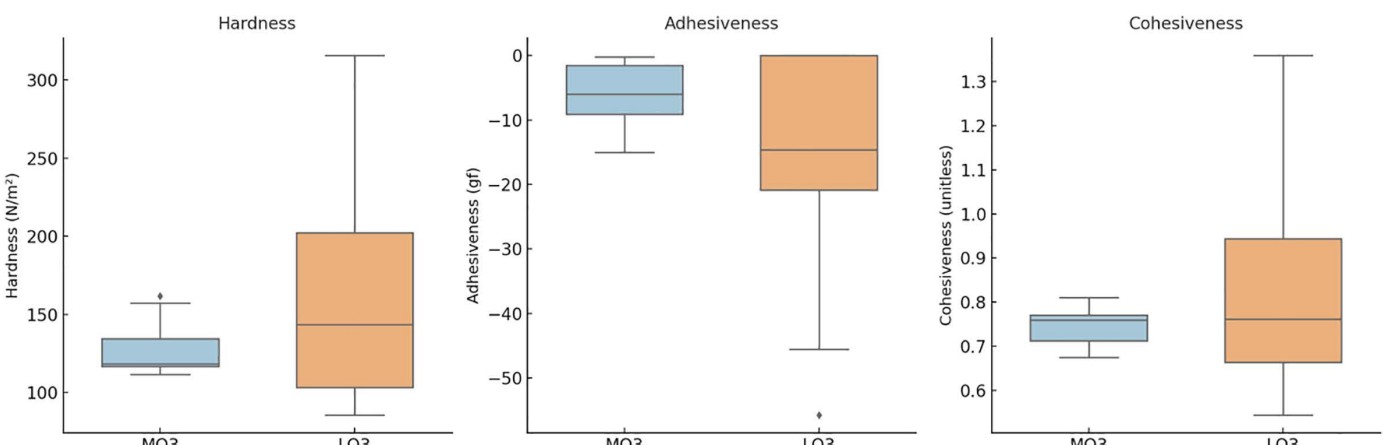

**Fig 2. Texture parameter comparison: Level 3 (MO3 vs LQ3).** Boxplots illustrating differences in hardness, adhesiveness, and cohesiveness between Level 3 subtypes. MO3 (liquid pathway) samples exhibited lower hardness and higher water content, whereas LQ3 (food pathway) showed greater structural density.

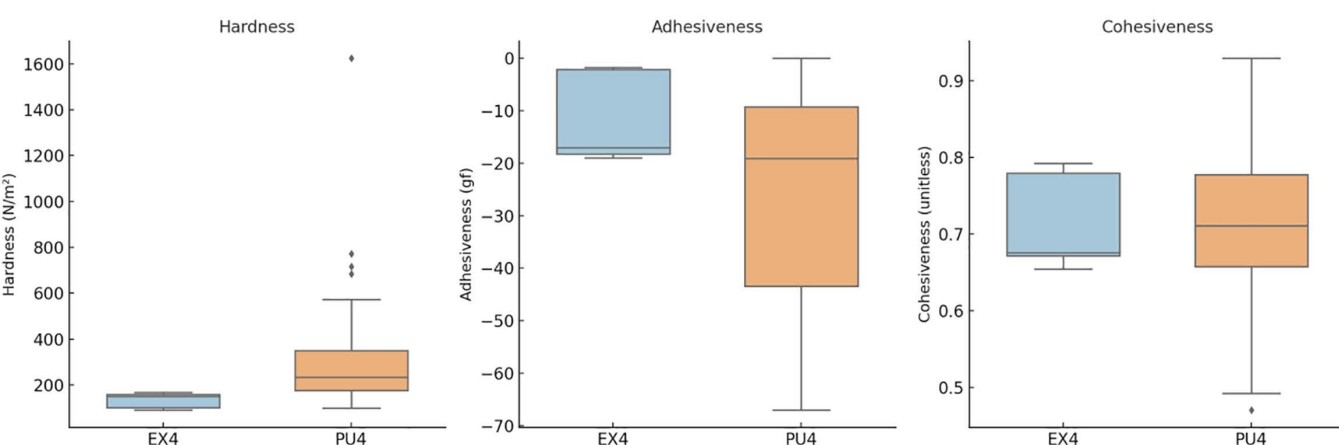

**Fig 3. Texture parameter comparison: Level 4 (EX4 vs PU4).** Boxplots comparing texture parameters (hardness, adhesiveness, cohesiveness) of Level 4 subtypes. PU4 (food pathway) exhibited higher hardness and nutrient density than EX4 (liquid pathway), which maintained higher moisture and lower viscosity.

 **Figures for section 3.2 — comparison between twin subtypes/ dual pathways.**

### 3.3 Ordinal trends within each path

 **3.3.1 Liquid Path (0 → 1 → 2 → MO3 → EX4).** A progressive, ordinal trend was observed along the liquid pathway, covering Levels 0 to EX4. Table 3 summarizes correlation coefficients (Spearman's ρ, Kendall's τ_b), q values (Benjamini–Hochberg FDR), and Theil–Sen slopes per level, while the overall monotonic trajectories are visualized in Fig 8.

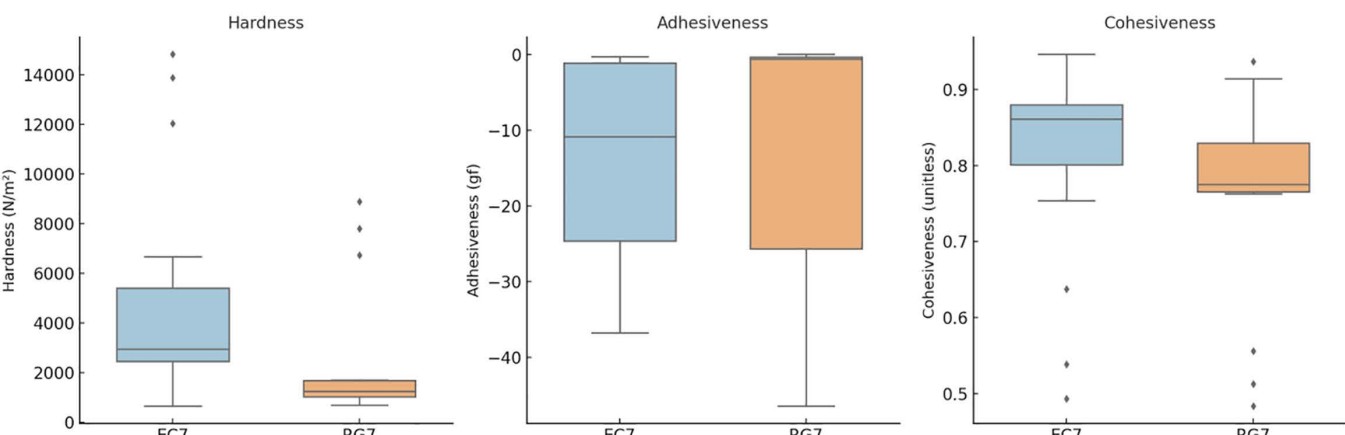

**Fig 4. Texture parameter comparison: Level 7 (EC7 vs RG7).** Boxplots showing the three texture indices across Level 7 subtypes. RG7 (regular foods) showed significantly higher hardness, while EC7 (easy-to-chew) maintained higher water content and softer texture, supporting safer oral processing.

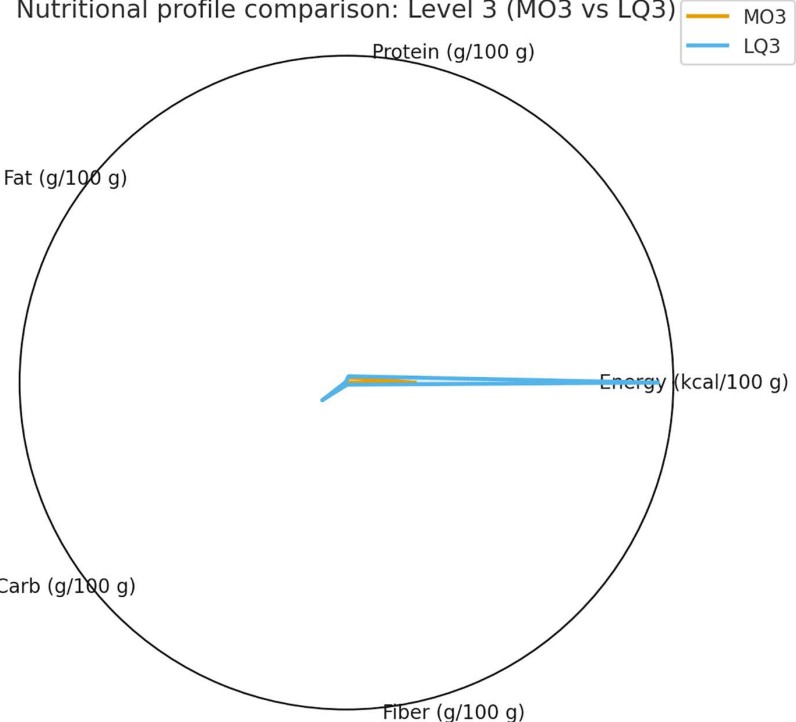

**Fig 5. Nutritional profile comparison: Level 3 (MO3 vs LQ3).** Radar chart comparing five nutritional indicators (energy, protein, fat, carbohydrate, fiber) between Level 3 subtypes. LQ3 presented markedly higher energy and macronutrient density than MO3.

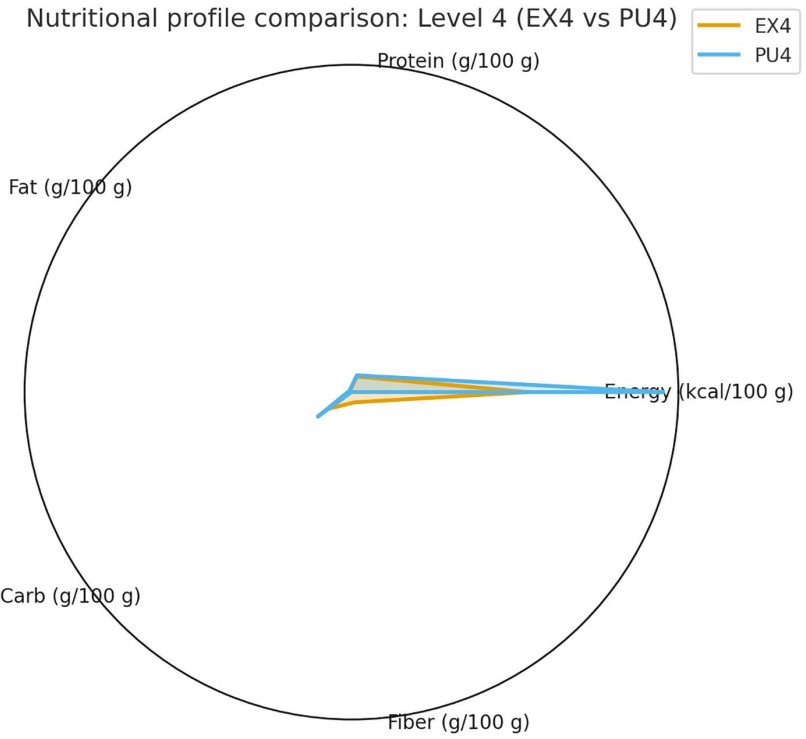

**Fig 6. Nutritional profile comparison: Level 4 (EX4 vs PU4).** Radar chart showing nutritional contrasts between Level 4 subtypes. PU4 demonstrated a more nutrient-dense composition, whereas EX4 emphasized higher water content and lower energy density.

In terms of texture, hardness increased significantly with ascending levels ($\rho=0.686$, $q<0.05$), with a Theil–Sen slope of approximately +9.87 N/m² per level.Adhesiveness showed a strong negative correlation ($\rho=-0.627$, $q<0.05$), while cohesiveness also declined ($\rho=-0.628$, $q<0.05$).For hydration-related indicators, water content per 100 g exhibited a decreasing trend with increasing level ($\rho=-0.192$, $q<0.05$), while the water ratio dropped even more markedly ($\rho=-0.385$, $q<0.05$; Theil–Sen $\approx-0.33$ per level). Median water content decreased from 84.35 g/100 g at Level 0 to approximately 150.79 g/100 g at EX4.Regarding nutritional indicators, a mild but consistent positive trend was detected for energy density ($\rho=0.159$, $q<0.05$; +0.50 kcal/100 g per level) and macronutrients. Protein ($\rho=0.291$, $q<0.05$; +0.06 g/100 g per level), fat ($\rho=0.243$, $q<0.05$; +0.03 g/100 g per level), and carbohydrate ($\rho=0.162$, $q<0.05$; +0.35 g/100 g per level) all increased progressively, whereas dietary fiber showed no significant association ($\rho=0.067$, $q>0.05$). Median energy rose from approximately 11.20 kcal/100 g at Level 0 to 40.00 kcal/100 g at EX4, and protein increased from 0.93 g/100 g to 1.20 g/100 g.

Overall, within the liquid path, hardness demonstrated a strong upward monotonic trend ($q\leq0.05$), adhesiveness decreased markedly, and hydration indices consistently declined. In contrast, energy, protein, fat, and carbohydrate contents increased with higher levels ($q\leq0.05$), while dietary fiber remained largely unchanged ($q>0.05$).

**3.3.2 Food path (LQ3 → PU4 → 5 → 6 → EC7 → RG7).** Along the food pathway, covering Levels LQ3 to RG7, texture and nutritional characteristics exhibited distinct ordinal trends (Table 4; Fig 9).

Hardness increased significantly with ascending levels (Spearman $\rho=0.808$, $q<0.05$), with a Theil–Sen slope of approximately +198 N/m² per level. The median hardness rose from 143.41 N/m² at LQ3 to 1242.23 N/m² at RG7. Adhesiveness ($\rho=-0.048$, $q>0.05$) and cohesiveness ($\rho=-0.055$, $q>0.05$) showed no significant monotonic trends.Water content decreased significantly with level advancement ($\rho=-0.277$, $q<0.05$; Theil–Sen $\approx-1.94$ g/100 g per level), dropping from 89.20 g/100 g at LQ3 to 75.20 g/100 g at RG7. Similarly, the water ratio exhibited a pronounced decline ($\rho=-0.500$,

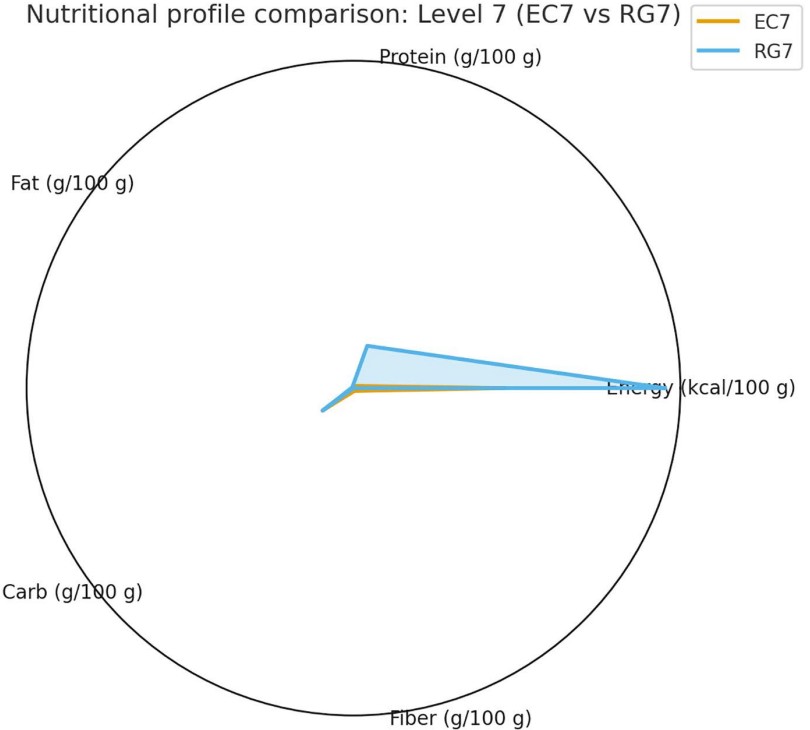

**Fig 7. Nutritional profile comparison: Level 7 (EC7 vs RG7).** Radar chart presenting the nutrient profiles of Level 7 subtypes. RG7 exhibited higher energy, protein, and fat contents, while EC7 retained higher moisture and easier chewability.

**Table 3. Spearman's ρ, _q_ values, Kendall's τ_b, and Theil–Sen slopes for indicators along the Liquid Path (0→1→2→MO3→EX4).**

| Metric | Spearman ρ | _q_ (BH-FDR) | Kendall τ_b | Theil–Sen slope/ level | Start→End (median [IQR]) |
|---|---|---|---|---|---|
| Hardness | 0.686 | <0.05 | 0.552 | +9.87 | 0: 84.35 [82.09–88.23] → EX4: 150.79 [101.73–…] |
| Adhesiveness | −0.627 | <0.05 | −0.544 | ~0 (downward) | 0: −6.00 [−9.10—1.49] → EX4: −17.65 [−22.21—12.60] |
| Cohesiveness | −0.628 | <0.05 | −0.494 | −0.111 | 0: 0.76 [0.71–0.77] → EX4: 0.80 [0.74–0.86] |
| Water content | −0.192 | <0.05 | −0.150 | −0.70 | 0: 84.35 [82.09–88.23] → EX4: 150.79 [101.73–197.77} |
| Water ratio | −0.385 | <0.05 | −0.321 | −0.33 | 0: 1.50 [1.50–2.00] → EX4: 0.50 [0.25–1.00] |
| Energy (kcal/100 g) | 0.159 | <0.05 | 0.122 | +0.50 | 0: 11.20 [9.33–42.00] → EX4: 40.00 [31.80–47.60] |
| Protein (g/100 g) | 0.291 | <0.05 | 0.244 | +0.06 | 0: 0.93 [0.67–1.13] → EX4: 1.20 [0.60–1.30] |
| Fat (g/100 g) | 0.243 | <0.05 | 0.195 | +0.03 | 0: 0.15 [0.12–0.16] → EX4: 0.20 [0.10–0.30] |
| Carbohydrate (g/100g) | 0.162 | <0.05 | 0.127 | +0.35 | 0: 2.00 [1.50–10.10] → EX4: 9.90 [8.40–12.80] |
| Dietary fiber (g/100g) | 0.067 | >0.05 | 0.065 | +0.02 | 0: 0.04 [0.00–0.62] → EX4: 1.00 [1.00–1.30] |

Significance judged at q≤0.05. EX4 water content >100 g/100 g suggests inclusion of free water during measurement; data reported as-is.

_q_ < 0.05; −0.25 per level), from 1.00 at LQ3 to 0.00 [0.00–0.00] at RG7.Energy content increased significantly with level (ρ = 0.223, _q_ < 0.05; +0.66 kcal/100 g per level), rising from 51.00 kcal/100 g at LQ3 to 106.00 kcal/100 g at RG7. Carbohydrates (ρ = 0.267, _q_ < 0.05; +1.05 g/100 g per level) and dietary fiber (ρ = 0.212, _q_ < 0.05; +0.02 g/100 g per level) also exhibited significant upward trends.

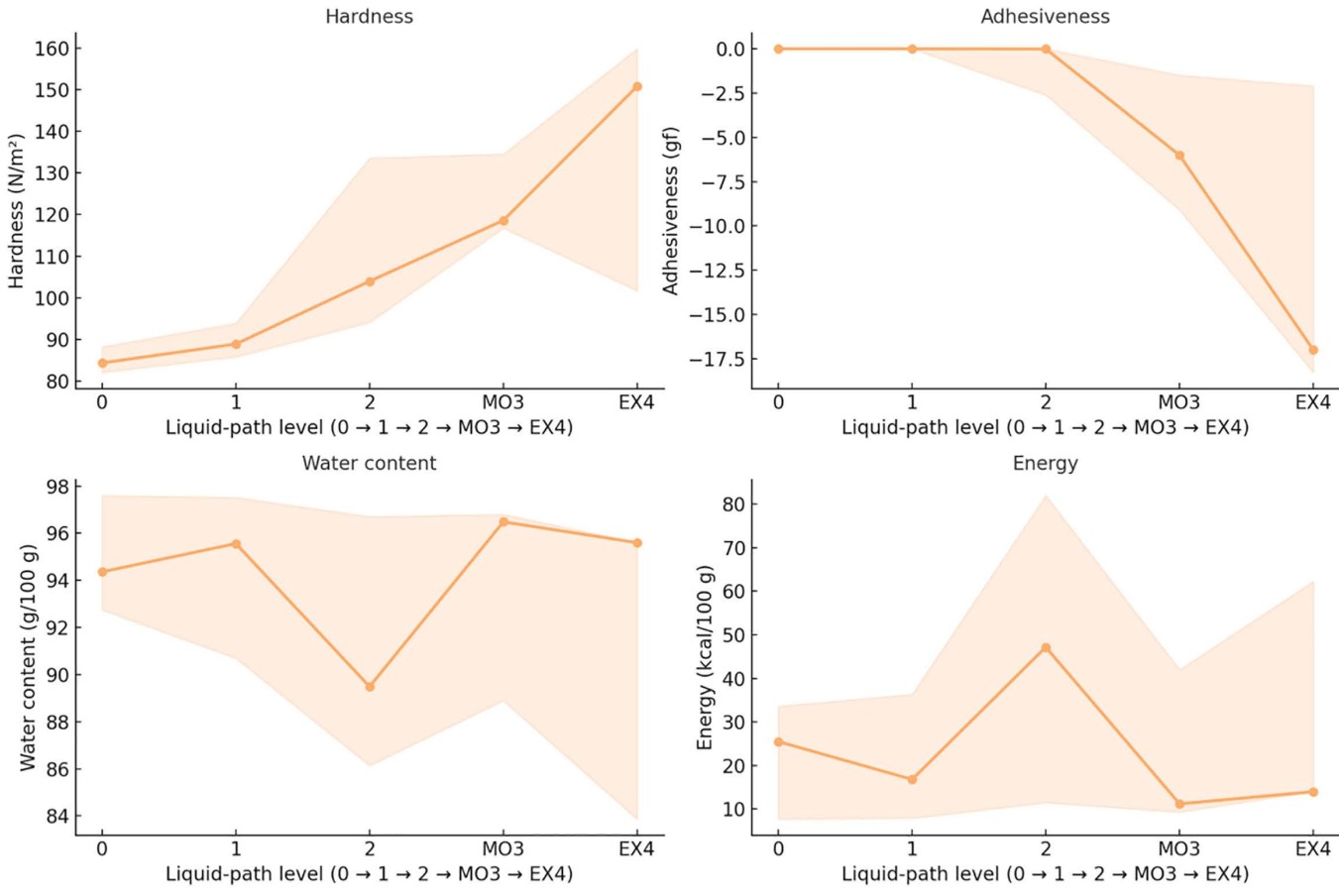

**Fig 8. Ordinal trend along the Liquid Path.** (Levels 0→1→2→MO3→EX4).

**Table 4. Spearman's ρ, q values, Kendall's τ_b, and Theil–Sen slopes for indicators along the Food Path (LQ3→PU4→5→6→EC7→RG7).**

| Metric | Spearman ρ | q (BH-FDR) | Kendall τ_b | Theil–Sen slope/ level | Start→End (median [IQR]) |
|---|---|---|---|---|---|
| Hardness | 0.808 | ≤0.05 | 0.659 | +198.11 | LQ3: 143.41 [103.40–202.37] → RG7: 1242.23 [1024.63–1418.22] |
| Adhesiveness | −0.048 | >0.05 | −0.025 | −0.094 | LQ3: −14.58 [−20.89–0.00] → RG7: −0.62 [−25.62–0.00] |
| Cohesiveness | −0.055 | >0.05 | −0.044 | −0.0064 | LQ3: 0.76 [0.66–0.94] → RG7: 0.78 [0.77–0.83] |
| Water content | −0.277 | ≤0.05 | −0.206 | −1.94 | LQ3: 89.20 [85.50–93.92] → RG7: 75.20 [51.50–78.10] |
| Water ratio | −0.500 | ≤0.05 | −0.413 | −0.25 | LQ3: 1.00 [0.50–1.50] → RG7: 0.00 [0.00–0.00] |
| Energy (kcal/100 g) | 0.223 | ≤0.05 | 0.167 | +0.66 | LQ3: 51.00 [22.80–71.50] → RG7: 106.00 [93.40–119.00] |
| Protein (g/100 g) | 0.109 | ≤0.05 | 0.083 | +0.125 | LQ3: 1.13 [0.73–4.45] → RG7: 15.20 [1.40–20.20] |
| Fat (g/100 g) | 0.119 | ≤0.05 | 0.087 | +0.010 | LQ3: 0.20 [0.10–0.40] → RG7: 0.60 [0.20–2.30] |
| Carbohydrate (g/100g) | 0.267 | ≤0.05 | 0.200 | +1.05 | LQ3: 4.96 [2.27–13.10] → RG7: 13.10 [3.40–22.00] |
| Dietary fiber (g/100g) | 0.212 | ≤0.05 | 0.168 | +0.02 | LQ3: 0.40 [0.00–1.00] → RG7: 0.10 [0.00–1.20] |

*Significance judged at q ≤ 0.05.*

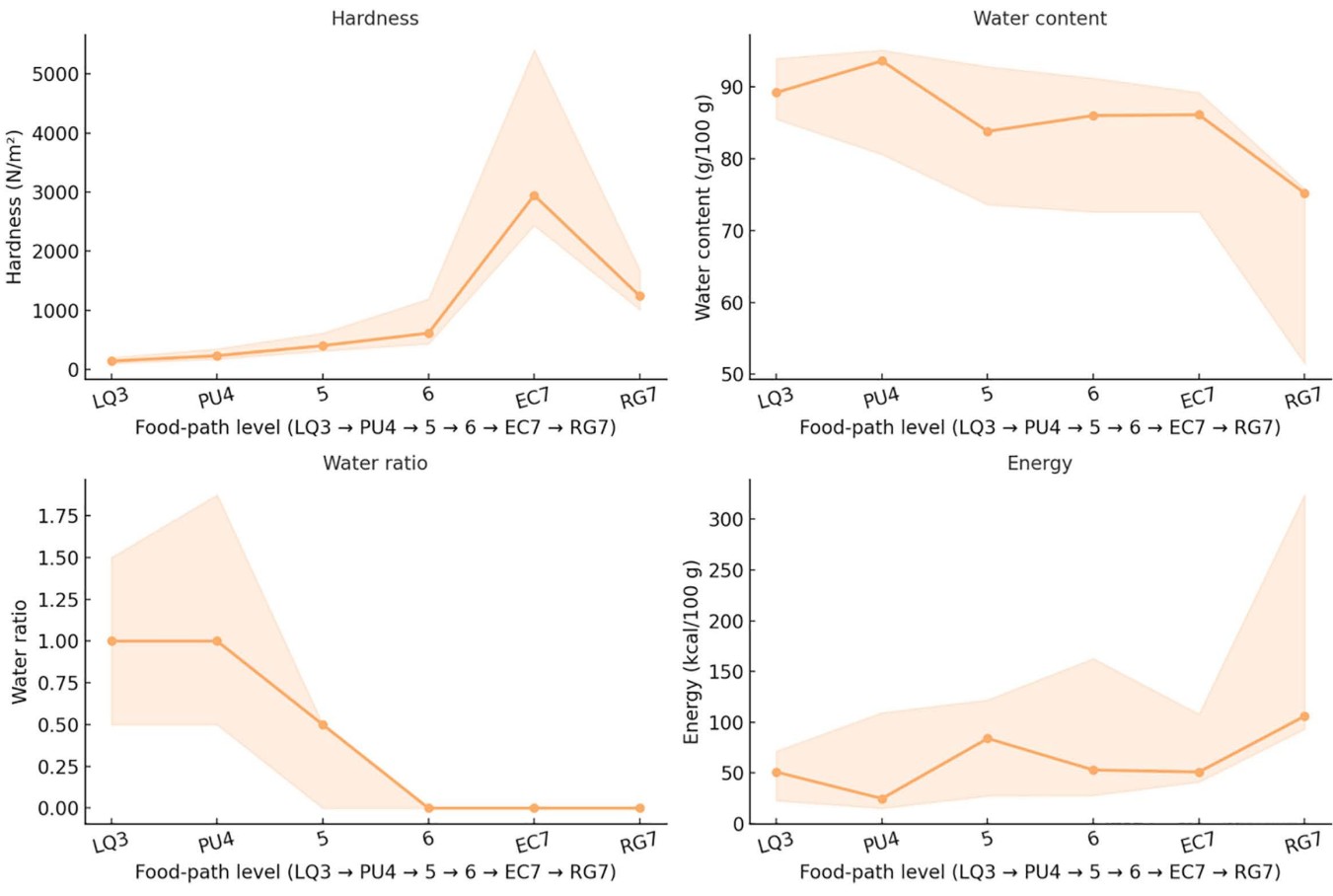

**Fig 9. Ordinal trend along the Food Path.** (LQ3→PU4→5→6→EC7→RG7).

Protein ($\rho=0.109$, $q=0.085$) and fat ($\rho=0.119$, $q=0.063$) showed weak positive associations that did not reach statistical significance after FDR correction.Nevertheless, endpoint comparisons revealed substantial increases: protein rose from 1.13 g/100 g in LQ3 to 15.20 g/100 g in RG7, and fat increased from 0.20 g/100 g to 0.60 g/100 g.

Within the food path, hardness showed a strong positive monotonic trend ($q\leq0.05$), while water content and water ratio decreased significantly ($q\leq0.05$). Energy, carbohydrate, and dietary fiber content increased progressively. Adhesiveness and cohesiveness did not vary significantly across levels ($q>0.05$).

### 3.4 Intra-level variability by food category

**Level 3 (MO3+LQ3 combined as "IDDSI Level 3").** Level 3 included 81 samples: grains and tubers (n = 24), vegetables (n = 24), fruit (n = 12), meat (n = 6), mixed beans (n = 6), other: tofu (n = 6), and egg (n = 3).Kruskal–Wallis tests showed significant differences across food categories for all indicators, with all results remaining significant after FDR correction ($q\leq0.05$): hardness (H = 35.77), adhesiveness (H = 38.99), cohesiveness (H = 33.65), water content (H = 52.02), energy (H = 54.96), and protein (H = 55.23). Effect sizes ($\varepsilon^2$) ranged from 0.37 to 0.67.Median hardness ranged from 108 N/m² in grains and tubers to 214 N/m² in other: tofu; egg showed 179.39 N/m². Adhesiveness varied across categories (e.g., fruit: –27.34 gf; grains and tubers: –1.33 gf; meat: 0.00 gf). Water content also differed (mixed beans: 67.72 g/100 g; grains and tubers: 92.17 g/100 g; vegetables: 89.20 g/100 g).Energy and protein content were higher in egg, meat, and mixed

beans than in fruit, vegetables, and grains and tubers (e.g., egg: 109.33 kcal/100 g and 6.84 g/100 g protein; fruit: 46.50 kcal/100 g and 0.50 g/100 g protein). A subtype × category χ² test indicated a non-random distribution of food categories between MO3 and LQ3.

**Level 4 (EX4＋PU4 combined as "IDDSI Level 4").** Level 4 included 55 samples: vegetables (n = 21), grains and tubers (n = 13), egg (n = 12), fruit (n = 3), meat (n = 3), and mixed beans (n = 3).All six indicators differed significantly across food categories (all $q \leq 0.001$): hardness (H = 24.74), adhesiveness (H = 22.88), cohesiveness (H = 22.47), water content (H = 31.12), energy (H = 38.22), and protein (H = 36.30). Effect sizes ($\varepsilon^2$) ranged from 0.36 to 0.68.Median hardness in egg (367.96 N/m²) exceeded fruit (98.36 N/m²) and meat (124.07 N/m²). Water content differed between categories (vegetables: 95.20 g/100 g; egg: 78.17 g/100 g; mixed beans: 75.16 g/100 g). Energy and protein were higher in egg and mixed beans than in vegetables and grains and tubers (egg: 147.60 kcal/100 g and 6.84 g/100 g protein; vegetables: 14.00 kcal/100 g and 1.10 g/100 g protein). A χ² test showed that food categories were not evenly distributed between EX4 and PU4.

**Level 7 (EC7＋RG7 combined as "IDDSI Level 7").** Level 7 included 30 samples: fruit (n = 12), grains and tubers (n = 9), vegetables (n = 6), egg (n = 3), meat (n = 3), and mixed beans (n = 3).Kruskal–Wallis tests were significant for all indicators ($q \leq 0.05$): hardness (H = 27.82, $\varepsilon^2$ = 0.76), adhesiveness (H = 14.75, $\varepsilon^2$ = 0.32), cohesiveness (H = 22.69, $\varepsilon^2$ = 0.59), water content (H = 28.33, $\varepsilon^2$ = 0.78), energy (H = 28.33, $\varepsilon^2$ = 0.78), and protein (H = 27.68, $\varepsilon^2$ = 0.76).Median hardness spanned widely across categories (egg: 744.05 N/m² [718.39–788.63]; meat: 1050.77 N/m²; grains and tubers: 1927.13 N/m² [1665.60–2723.51]; fruit: 6699.57 N/m²). Water content also differed (vegetables: 92.20 g/100 g [89.20–95.20]; fruit: 86.00 g/100 g; mixed beans: 12.60 g/100 g).Energy and protein content were highest in egg and mixed beans (egg: 328.00 kcal/100 g and 15.20 g/100 g protein; mixed beans: 324.00 kcal/100 g and 20.20 g/100 g protein) and lower in fruit (52.00 kcal/100 g and 0.50 g/100 g protein). A χ² test indicated non-random distribution of food categories between EC7 and RG7.

Tables 5–7. Within-level variability by food category across IDDSI Level 3, Level 4, and Level 7, including sample size per category, median [IQR] for hardness, adhesiveness, cohesiveness, water content, energy, and protein, and Kruskal–Wallis results ($\varepsilon^2$ and FDR-adjusted $q$ values).

**Table 5. Within-level variability by food category in IDDSI Level 3 (MO3＋LQ3 combined).**

| Food category | n | Hardness (N/m²) median [IQR] | Adhesiveness (gf) median [IQR] | Cohesiveness median [IQR] | Water content (g/100 g) median [IQR] | Energy (kcal/100 g) median [IQR] | Protein (g/100 g) median [IQR] |
|---|---|---|---|---|---|---|---|
| Egg | 3 | 179.39 [159.03–211.92] | −15.59 [−17.00—15.07] | 0.71 [0.68–0.71] | 83.83 [83.83–83.83] | 109.33 [109.33–109.33] | 6.84 [5.83–8.23] |
| Fruit | 12 | 156.03 [132.85–225.52] | −27.34 [−32.95—18.85] | 0.75 [0.73–0.76] | 87.50 [86.05–88.90] | 46.50 [42.00–51.50] | 0.50 [0.50–0.50] |
| Grains and tubers | 24 | 108.00 [96.10–127.73] | −1.33 [−8.95–0.00] | 0.87 [0.77–1.00] | 92.17 [62.20–93.36] | 30.57 [26.41–146.23] | 0.95 [0.52–1.40] |
| Meat | 6 | 109.75 [103.41–114.70] | 0.00 [−0.08–0.00] | 0.87 [0.81–0.95] | 86.15 [85.50–86.80] | 61.50 [51.50–71.50] | 5.05 [5.05–5.05] |
| Mixed beans | 6 | 161.60 [116.89–207.31] | −7.48 [−15.70—0.23] | 0.67 [0.57–0.77] | 67.72 [56.15–79.30] | 122.91 [81.33–164.50] | 4.45 [4.45–4.45] |
| Other: tofu | 6 | 214.48 [176.57–249.73] | −20.48 [−22.11—16.95] | 0.61 [0.60–0.63] | 89.20 [89.20–89.20] | 56.00 [56.00–56.00] | 2.80 [2.80–2.80] |
| Vegetable | 24 | 138.11 [118.38–162.96] | −6.46 [−14.15—1.03] | 0.76 [0.74–0.90] | 89.20 [86.50–91.95] | 56.00 [56.00–56.00] | 1.10 [1.00–1.40] |

Kruskal–Wallis, FDR-adjusted $q \leq 0.05$.

**Table 6. Within-level variability by food category in IDDSI Level 4 (EX4＋PU4 combined).**

| Food category | n | Hardness (N/m²) median [IQR] | Adhesiveness (gf) median [IQR] | Cohesiveness median [IQR] | Water content (g/100 g) median [IQR] | Energy (kcal/100 g) median [IQR] | Protein (g/100 g) median [IQR] |
|---|---|---|---|---|---|---|---|
| Egg | 12 | 367.96 [254.92–605.40] | −12.65 [−17.07—5.88] | 0.69 [0.56–0.75] | 78.17 [73.73–81.41] | 147.60 [125.73–177.67] | 6.84 [5.83–8.23] |
| Fruit | 3 | 98.36 [94.99–100.04] | −1.77 [−1.94—0.91] | 0.79 [0.79–0.86] | 83.87 [83.87–83.87] | 62.27 [62.27–62.27] | 0.93 [0.93–0.93] |
| Grains and tubers | 13 | 212.78 [176.16–340.47] | −31.37 [−45.39—19.66] | 0.69 [0.66–0.78] | 95.10 [90.13–95.10] | 19.67 [19.67–39.89] | 0.47 [0.43–1.38] |
| Meat | 3 | 124.07 [120.32–125.33] | 0.00 [0.00–0.00] | 0.90 [0.89–0.90] | 93.80 [93.80–93.80] | 26.50 [26.50–26.50] | 5.05 [5.05–5.05] |
| Mixed beans | 3 | 299.09 [285.98–302.44] | −26.94 [−27.45—24.73] | 0.51 [0.50–0.51] | 75.16 [75.16–75.16] | 97.60 [97.60–97.60] | 1.80 [1.80–1.80] |
| Vegetable | 21 | 183.17 [169.38–297.13] | −19.02 [−53.13—10.13] | 0.70 [0.68–0.79] | 95.20 [94.60–95.80] | 14.00 [13.50–20.60] | 1.10 [0.90–1.40] |

Kruskal–Wallis, FDR-adjusted $q \le 0.05$.

**Table 7. Within-level variability by food category in IDDSI Level 7 (EC7＋RG7 combined).**

| Food category | n | Hardness (N/m²) median [IQR] | Adhesiveness (gf) median [IQR] | Cohesiveness median [IQR] | Water content (g/100 g) median [IQR] | Energy (kcal/100 g) median [IQR] | Protein (g/100 g) median [IQR] |
|---|---|---|---|---|---|---|---|
| Egg | 3 | 744.05 [718.39–788.63] | −0.42 [−0.83—0.41] | 0.91 [0.90–0.93] | 51.50 [51.50–51.50] | 328.00 [328.00–328.00] | 15.20 [15.20–15.20] |
| Fruit | 12 | 6699.57 [5299.72–9671.19] | −20.44 [−36.53—2.25] | 0.87 [0.83–0.89] | 86.00 [83.38–86.80] | 52.00 [48.75–63.10] | 0.50 [0.38–0.80] |
| Grains and tubers | 9 | 1927.13 [1665.60–2723.51] | −16.06 [−25.42—10.88] | 0.84 [0.80–0.86] | 72.60 [5.60–85.30] | 108.00 [59.00–362.00] | 1.40 [1.30–8.40] |
| Meat | 3 | 1050.77 [1013.99–1055.59] | −0.22 [−0.28—0.11] | 0.77 [0.77–0.77] | 75.20 [75.20–75.20] | 106.00 [106.00–106.00] | 20.20 [20.20–20.20] |
| Mixed beans | 3 | 1242.23 [1218.67–1471.55] | −0.54 [−0.58—0.34] | 0.51 [0.50–0.53] | 12.60 [12.60–12.60] | 324.00 [324.00–324.00] | 20.20 [20.20–20.20] |
| Vegetable | 6 | 1649.70 [772.54–2816.04] | −5.99 [−21.34—0.43] | 0.70 [0.56–0.79] | 92.20 [89.20–95.20] | 28.10 [15.00–41.20] | 0.95 [0.90–1.00] |

Kruskal–Wallis, FDR-adjusted $q \le 0.05$.

### 3.5 Safety–nutrition trade-off windows

Table 8 summarizes the empirical distribution windows of key indicators—hardness, adhesiveness, water content, energy, and protein—across representative IDDSI subtypes (MO3, EX4, PU4, EC7, RG7).

For each subtype, the interquartile range (P25–P75) represents the target operational band, and the P10–P90 interval represents the release tolerance range.

In MO3, hardness ranged from 116.76–134.53 N/m² (P25–P75), adhesiveness from −9.10 to −1.49 gf, and water content from 88.90–96.80 g/100 g. Energy and protein densities were 9.33–42.00 kcal/100 g and 0.67–1.13 g/100 g, respectively.

EX4 showed hardness of 101.73–159.82 N/m², adhesiveness −18.27 to −2.10 gf, and water content 88.90–94.80 g/100 g; energy ranged from 14.00–62.27 kcal/100 g, and protein 0.93–1.30 g/100 g.

PU4 presented hardness of 176.67–350.30 N/m², adhesiveness −43.44 to −9.20 gf, water content 75.16–95.10 g/100 g, energy 15.33–109.33 kcal/100 g, and protein 0.91–5.05 g/100 g.

**Table 8. Suggested operational windows for key IDDSI subtypes (*P25–P75 = target band; P10–P90 = release tolerance band*).**

| Subtype (n) | Parameter | P25–P75 (Target range) | P10–P90 (Release range) |
|---|---|---|---|
| MO3 (n = 24) | Hardness (N/m²) | 116.76–134.53 | 114.27–151.71 |
| | Adhesiveness (gf) | −9.10 – −1.49 | −11.98 – −0.35 |
| | Water content (g/100 g) | 88.90–96.80 | 79.30–97.40 |
| | Energy (kcal/100 g) | 9.33–42.00 | 7.00–81.33 |
| | Protein (g/100 g) | 0.67–1.13 | 0.60–4.60 |
| EX4 (n = 5) | Hardness (N/m²) | 101.73–159.82 | 95.67–164.35 |
| | Adhesiveness (gf) | −18.27 – −2.10 | −18.72 – −1.90 |
| | Water content (g/100 g) | 88.90–94.80 | 88.90–95.20 |
| | Energy (kcal/100 g) | 14.00–62.27 | 14.00–62.27 |
| | Protein (g/100 g) | 0.93–1.30 | 0.93–1.30 |
| PU4 (n = 50) | Hardness (N/m²) | 176.67–350.30 | 160.33–568.84 |
| | Adhesiveness (gf) | −43.44 – −9.20 | −55.63 – −2.19 |
| | Water content (g/100 g) | 75.16–95.10 | 75.16–95.20 |
| | Energy (kcal/100 g) | 15.33–109.33 | 13.50–181.00 |
| | Protein (g/100 g) | 0.91–5.05 | 0.47–7.60 |
| EC7 (n = 21) | Hardness (N/m²) | 2438.74–5400.02 | 860.65–12042.51 |
| | Adhesiveness (gf) | −24.63 – −1.14 | −28.94 – −0.30 |
| | Water content (g/100 g) | 72.60–89.20 | 5.60–95.20 |
| | Energy (kcal/100 g) | 41.20–108.00 | 15.00–362.00 |
| | Protein (g/100 g) | 0.40–1.40 | 0.30–8.40 |
| RG7 (n = 15) | Hardness (N/m²) | 1013.99–1683.24 | 779.72–7376.46 |
| | Adhesiveness (gf) | −25.62 – −0.37 | −38.77 – −0.18 |
| | Water content (g/100 g) | 51.50–75.80 | 12.60–85.30 |
| | Energy (kcal/100 g) | 93.40–324.00 | 59.00–328.00 |
| | Protein (g/100 g) | 1.40–20.20 | 1.30–20.20 |

EC7 showed hardness of 2438.74–5400.02 N/m², adhesiveness −24.63 to −1.14 gf, and water content 72.60–89.20 g/100 g. Energy ranged 41.20–108.00 kcal/100 g, and protein 0.40–1.40 g/100 g.

RG7 exhibited hardness of 1013.99–1683.24 N/m², adhesiveness −25.62 to −0.37 gf, water content 51.50–75.80 g/100 g, energy 93.40–324.00 kcal/100 g, and protein 1.40–20.20 g/100 g.

Overall, the data indicate a gradual increase in hardness and nutritional density, accompanied by a decline in hydration indicators, from liquid-path (MO3–EX4) to food-path (PU4–RG7) subtypes.

## 4. Discussion

### 4.1 Mechanistic explanation and constitutive control of intra-level differentiation

This study systematically compared dual pathway subtypes (MO3 vs. LQ3, EX4 vs. PU4, EC7 vs. RG7) within IDDSI Levels 3, 4, and 7. The results consistently indicate that liquid-path subtypes (MO3, EX4) generally exhibit lower hardness, lower adhesive burden, and higher water content compared with food-path subtypes (LQ3, PU4), while food-path subtypes demonstrate significantly higher energy and protein densities. At Level 7, EC7 shows lower hardness than RG7, reflecting a reduced chewing threshold and improved oral breakdown properties [5,6].Importantly, IDDSI levels are defined primarily by functional swallowing performance tests rather than by instrumental texture parameters alone [5,7]. Therefore, the "twin subtype" differentiation observed here should not be interpreted as redefining IDDSI levels but rather as describing objective variability within

a shared functional classification. The drink pathway emphasizes controlled bolus flow, airway protection, and swallow initiation safety, whereas the food pathway prioritizes bolus cohesion, oral manipulation, and controlled breakdown during mastication [7,13]. Instrumental texture profile analysis (TPA) metrics such as hardness, adhesiveness, and cohesiveness thus serve as quantitative descriptors supporting — but not replacing — established IDDSI bedside tests [10,13].These findings suggest that intra-level variability reflects differences in swallowing functional targets rather than inconsistencies in classification. Substituting foods solely based on numeric IDDSI level without considering pathway characteristics may therefore lead to mismatches in hydration adequacy, aspiration risk, and nutritional delivery. Previous studies have similarly reported trade-offs between thickening, aspiration prevention, and post-swallow residue formation [12,14]. Consequently, the present results suggest that considering pathway characteristics alongside numeric IDDSI levels may help inform dysphagia diet planning and texture management, although clinical validation remains necessary.

### 4.2 Ordinal trends within pathways and functional trajectories

The study identified a continuous "thickening–hardening–dehydrating" trajectory along the liquid pathway (Levels 0–4), characterized by progressive increases in hardness and adhesiveness and concurrent reductions in cohesiveness and water content as IDDSI level increased. Nutritional parameters (energy, protein, fat, and carbohydrates) showed weak-to-moderate positive correlations with level progression ($q \leq 0.05$), suggesting that nutrient density is influenced primarily by formulation strategies rather than texture consistency itself [15,16]. In contrast, the food pathway (Levels 3–7) exhibited a "structural hardening–dehydrating" trajectory, with a pronounced increase in hardness and a significant reduction in water content ($q \leq 0.05$). However, adhesiveness and cohesiveness did not show consistent monotonic trends, likely reflecting the combined influence of particle size, structural breakdown characteristics, porosity, and surface interactions on oral–pharyngeal bolus transport at higher IDDSI levels [9,11,12,17].These pathway-specific patterns are broadly consistent with previous reports describing trade-offs between thickening, aspiration mitigation, post-swallow residue, and oral processing demands, as well as the role of particle size in swallowing biomechanics [12,14]. Rather than defining clinical phenotypes directly, the present findings may help explain commonly observed differences in tolerance between liquid and solid dysphagia diets, although direct clinical validation remains necessary.Furthermore, prior studies have demonstrated that nutritional fortification strategies can improve energy and protein intake when texture safety is preserved. Our results extend this literature by quantitatively illustrating how instrumental texture parameters, hydration status, and nutrient density interact within the same functional IDDSI classification, thereby providing a potential framework for balancing swallowing safety, hydration adequacy, and nutritional sufficiency in texture-modified diets.

### 4.3 Clinical application transformation: pathway-oriented texture management and substitution considerations

The present findings provide potential insights for dysphagia diet management rather than prescriptive clinical recommendations. For individuals experiencing greater difficulty with liquid swallowing (e.g., choking tendency, delayed swallow initiation, or increased hydration and medication needs), progression within the liquid-oriented pathway (MO3→EX4) may help maintain hydration while preserving manageable bolus flow characteristics. The relatively lower hardness, reduced adhesiveness, and higher water content observed along this pathway may contribute to swallow facilitation, although direct clinical validation remains necessary. Nutritional adequacy in such contexts may require targeted formula enrichment to compensate for dilution effects [15].

Conversely, individuals with greater challenges in chewing or oral bolus formation but relatively preserved liquid swallowing may benefit from gradual progression within the food-oriented pathway (LQ3→PU4→EC7→RG7). Texture adjustments along this pathway primarily involve particle size control, structural breakdown properties, and hydration balance. Excessively abrupt increases in hardness or reductions in water content could potentially increase oral fatigue or residue risk, emphasizing the importance of gradual texture transitions [17]. The EC7 subtype at Level 7, characterized by reduced bite and crumble thresholds, may represent an intermediate structural stage facilitating oral processing [6].

Rather than advocating strict substitution rules, the present results highlight the importance of considering pathway characteristics alongside numeric IDDSI levels when planning dysphagia diets. Instrumental texture profile analysis (TPA) metrics, hydration status, and nutrient density may serve as complementary quantitative indicators in addition to established IDDSI bedside functional tests [5,10,13].Adjusting one major dimension at a time (e.g., hydration level or particle size before nutritional fortification) may help maintain balance between swallowing safety, hydration adequacy, and nutritional sufficiency, although individualized clinical monitoring remains essential.

Overall, these observations suggest that pathway-oriented texture management may provide a useful conceptual framework for optimizing dysphagia diet formulation while respecting the primacy of functional swallowing assessment in clinical decision-making.

### 4.4  Safety-nutrition trade-off windows: translating data into clinical quality control

In order to translate the results into actionable quality control indicators, this study identified the operational windows for key IDDSI subtypes (MO3, EX4, PU4, EC7, RG7) based on the empirical distributions of hardness, adhesiveness, water content, energy density, and protein density. For each subtype, the 25th–75th percentile range (target range) and the 10th–90th percentile range (release tolerance range) were calculated. Tables 3–5 summarizes these operational windows, providing clear targets for clinical prescription and substitution boundaries.

The data reveal that Liquid-path subtypes (MO3, EX4) tend to fall within a "low hardness, high water, moderate-low energy/protein" window, while Food-path and chewing-end subtypes (PU4, EC7, RG7) progressively enter a "higher hardness, lower water, higher energy/protein" window [13,15]. Notably, even within the same numeric IDDSI level (e.g., EX4 vs. PU4 at Level 4, EC7 vs. RG7 at Level 7), the target ranges for these subtypes do not overlap, indicating that cross-path substitution is not equivalent [13]. This reinforces the necessity for pathway-priority approaches in clinical practice, especially for patients with liquid or solid swallowing difficulties [7]. The operational windows presented here offer a minimum standard for clinical practitioners to guide food texture adjustments and recipe formulations, ensuring a balance between safety, hydration, and nutrition [10,13,16].

## 5.  Conclusion

This study demonstrates that within the same IDDSI functional level, objectively distinguishable texture–nutrition subtypes exist, characterized by differences in mechanical properties, hydration profiles, and nutrient density. These findings suggest that IDDSI classification alone may not fully capture intra-level variability relevant to clinical nutrition management.

The identification of hydration-oriented liquid pathways and structure-oriented food pathways provides a practical framework for optimizing dysphagia diets. By considering both functional swallowing requirements and nutritional targets simultaneously, clinicians and caregivers may better balance swallowing safety, hydration adequacy, and energy–protein delivery, thereby supporting improved nutritional status among individuals with dysphagia.

Although the present findings are derived from standardized food-ingredient models rather than direct clinical swallowing validation, they provide quantitative reference ranges that may inform dysphagia diet formulation, nutritional fortification strategies, and texture-prescription refinement. Future clinical translation would benefit from studies incorporating videofluoroscopic swallowing study (VFSS) and fiberoptic endoscopic evaluation of swallowing (FEES), which are imaging-based examinations used to assess swallowing physiology and aspiration risk, together with clinical outcome measures.

Overall, this study contributes objective evidence for integrating instrumental texture metrics with nutritional considerations within the IDDSI framework and supports a more nuanced, pathway-oriented approach to dysphagia nutrition care.

## Supporting information

**S1 Data. Raw data underlying all analyses in this study.**
(XLSX)

## Author contributions

**Conceptualization:** Muxi Chen, Yi Cheng.

**Data curation:** Muxi Chen, Juan Duan.

**Formal analysis:** Muxi Chen, Juan Duan.

**Funding acquisition:** Lei Shi, Yuan Liu.

**Investigation:** Muxi Chen, Yi Cheng.

**Methodology:** Muxi Chen.

**Resources:** Lei Shi, Yuan Liu.

**Software:** Muxi Chen.

**Supervision:** Lei Shi, Yuan Liu.

**Validation:** Muxi Chen.

**Visualization:** Muxi Chen, Juan Duan.

**Writing – original draft:** Muxi Chen, Yi Cheng.

**Writing – review & editing:** Yuan Liu.

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
