## [Decision Letter · Decision Letter 0]

26 Jan 2026

Dear Dr. Chen,

Thank you for submitting your manuscript to PLOS ONE. After careful consideration, we feel that it has merit but does not fully meet PLOS ONE’s publication criteria as it currently stands. Therefore, we invite you to submit a revised version of the manuscript that addresses the points raised during the review process.

We look forward to receiving your revised manuscript.

Kind regards,

António Raposo

Academic Editor

PLOS One

Journal Requirements:

2. In your Methods section, please include additional information about your dataset and ensure that you have included a statement specifying whether the collection and analysis method complied with the terms and conditions for the source of the data.

“Funding：Y.L. received funding from the project "Key Technologies and New Product Development for Intelligent and Safe Production of Foods for Special Medical Purposes" (Grant No. JH2019045).L.S. received funding from the project "Development and Industrialization Demonstration of Nutrition and Health Foods for the Elderly" (Grant No. 2023YFF1104405), funded by the Ministry of Science and Technology of China.”

5. We note that your Data Availability Statement is currently as follows: All relevant data are within the manuscript and in Supporting Information files.

7. We note that Figure 2 in your submission contain copyrighted images. All PLOS content is published under the Creative Commons Attribution License (CC BY 4.0), which means that the manuscript, images, and Supporting Information files will be freely available online, and any third party is permitted to access, download, copy, distribute, and use these materials in any way, even commercially, with proper attribution. For more information, see our copyright guidelines: http://journals.plos.org/plosone/s/licenses-and-copyright.

1. You may seek permission from the original copyright holder of Figure 2 to publish the content specifically under the CC BY 4.0 license.

Reviewers' comments:

Reviewer's Responses to Questions

**Comments to the Author**

1. Is the manuscript technically sound, and do the data support the conclusions?

Reviewer #1: Partly

Reviewer #2: Yes

2. Has the statistical analysis been performed appropriately and rigorously?

Reviewer #1: Yes

Reviewer #2: Yes

3. Have the authors made all data underlying the findings in their manuscript fully available?

Reviewer #1: Yes

Reviewer #2: Yes

4. Is the manuscript presented in an intelligible fashion and written in standard English?

Reviewer #1: No

Reviewer #2: Yes

Reviewer #1: The article attempts to pioneer the study on translating IDDSI to texture, prescription boundary system for dysphagia. The authors should: 1) that a better understanding is ensured especially in the discussion section 2) In the concluding section, one or two sentences on how nutrition is enhanced among those with dysphagia as a result of this study 3) What are the limitations of the present study 4) The authors recommended for future studies, the use of VFSS/FEES, it should be briefly mentioned that they are imaging tests for diagnosing swallowing disorders (dyspagia), also give their full meaning, e.g. VFSS, videofluoroscopic swallowing study 5) In many cases, the word ingredients was used, it is better to adopt food-ingredients in all cases 6) In the attached manuscript, comment-bubbles were highlighted for the authors' attention 7) Table 3-2 can be improved by writing those levels - Level 3 (MO3 vs LQ3), Level 4 (EX4 vs PU4), Level 7 (EC7 vs RG7) in bold 8) The manuscript will benefit from a few more relevant citations.

Reviewer #2: The topic is timely and clinically relevant: using IDDSI with objective texture metrics plus nutrition, and explicitly interrogating “twin” subtypes (e.g., MO3 vs LQ3) is a useful framing.

IDDSI Levels are defined by functional tests (flow test, fork drip, spoon tilt, fork pressure, etc.), not by the “pathway” label alone. You treat MO3 vs LQ3, EX4 vs PU4, EC7 vs RG7 as “same level” twins, but it’s not clear these are actually IDDSI-equivalent items with a meaningful shared level, versus two different categories that happen to share a number.

You need to explicitly define what MO/LQ/EX/PU/EC/RG mean (they look like internal abbreviations) and justify why they are “same level” rather than simply different IDDSI categories.

Provide a clear mapping table from your subtype labels to official IDDSI terminology and tests, and demonstrate that each sample passes the appropriate IDDSI test(s) for its assigned level.

“Python-based web crawling” to collect “ingredients” is vague. Are these raw foods, cooked foods, prepared dishes, commercial products, or database entries? If these are only nutrient profiles scraped from the web, then texture testing on physical samples becomes unclear.

“Commonly consumed in China” plus “elderly-preferred” requires evidence: how were these ingredients selected, and by whom? Without a sampling frame, this risks convenience sampling dressed as representativeness.

Regional coverage (“seven major dietary regions”) sounds good, but you don’t describe how many items per region, whether items overlap, and whether region is treated as a factor.

Provide full TPA settings, sample prep standardisation (including water content control), temperature control, number of replicates, and whether you used IDDSI reference methods alongside instrument tests.

Nutrition estimation method is a major validity bottleneck

“Calculated using Huaxi Hospital Nutrition software” implies a food composition database approach. That is not equivalent to chemical analysis and can be unreliable for prepared foods, thickened liquids, or modified textures where water addition is substantial.

Reporting per 100 g while comparing liquids vs foods can be misleading because dilution drives apparent nutrient density. Your “liquid pathway emphasizes hydration” conclusion could be an artefact of higher water content rather than a meaningful “orientation.”

At minimum, report nutrients per 100 mL for liquids, per serving, and consider energy/protein density (e.g., kcal/mL, g protein/100 kcal). If possible, measure moisture content directly and model nutrient density adjusted for water fraction.

Reframe conclusions: “potential implications” rather than “confirms safety.” Add a validation plan linking instrument metrics to IDDSI tests and then to clinical outcomes.

he premise is strong, but you need (1) unambiguous definitions and IDDSI compliance verification, (2) rigorous, reproducible sample preparation and texture protocol, (3) nutrition metrics that account for dilution/moisture and serving basis, and (4) more cautious, evidence-aligned claims about “safety” and clinical substitution.

.

Reviewer #1: No

Reviewer #2: **Yes:** M. João LimaM. João LimaM. João LimaM. João Lima

---

## [Author Response · Author response to Decision Letter 1]

11 Feb 2026

Response to Reviewers

Manuscript ID: PONE-D-25-62691

Title: Objective Differences and Pathway Differentiation between Twin Subtypes within the Same IDDSI Level: A Texture–Nutrition Analysis Based on Commonly Consumed Foods among Chinese Elderly

Dear Academic Editor and Reviewers,

We sincerely thank the Academic Editor and both reviewers for their thoughtful and constructive feedback. We have carefully revised the manuscript to address all of the points raised and believe that the revisions have significantly improved the clarity, transparency, and overall quality of the manuscript. Below, we provide detailed responses to each comment raised by the reviewers, as well as a summary of the changes made to the manuscript. All revisions have been incorporated into the manuscript and are marked in the tracked-changes version.

Responses to Journal Requirements

1. PLOS ONE formatting requirements

Response:

We have ensured that the manuscript fully complies with the PLOS ONE formatting guidelines, including file naming, title page, section structure, figure presentation, and reference style. The manuscript has been reformatted accordingly, and we will ensure that all files follow PLOS ONE’s file-naming conventions during submission.

Revision made:

• Formatting and structure updates applied throughout the manuscript.

2. Dataset description and compliance with data-source terms

Response:

We have added detailed information in the Methods section to clarify the dataset used in the study. Specifically, we clarified that web-scraped data were used only for ingredient selection and dietary context, not for experimental measurements. We also confirmed that the data collection process adhered to the terms and conditions of the source data.

Revision made:

• Methods Section 2.1 ("Ingredient Data Collection") has been updated to specify that web-derived information was used solely for ingredient selection and dietary context, and all experimental texture and nutritional data were collected from laboratory-prepared samples.

3. Funding information removal from the manuscript

Response:

As requested, we have removed all funding-related information from the manuscript body. The funding information has been provided exclusively in the online submission system’s Funding Statement section in compliance with PLOS ONE’s policy.

Revision made:

• Removed all funding-related text from the manuscript body.

4. Role of funders

Response:

We confirm that the funder (who is also the corresponding author) provided academic supervision and final approval of the manuscript but had no role in study design, data collection and analysis, or decision to publish. This clarification has been included in the cover letter and in the online submission form.

Revision made:

• Cover letter includes a statement on the funders' role:

"The funder had no role in study design, data collection and analysis, decision to publish, or preparation of the manuscript. The funder provided academic supervision and final approval of the manuscript."

5. Data availability / minimal dataset

Response:

We confirm that the minimal dataset necessary to replicate all reported findings has been provided. This includes the raw data underlying the summary statistics, the data points used to create graphs, and any metadata. The data are available as Supporting Information files.

Revision made:

• Data Availability Statement updated to indicate that the minimal dataset is included as Supporting Information, and we clarified the full accessibility of the data.

6. Data sharing plan

Response:

We acknowledge PLOS ONE’s open data policy and confirm that all data underlying the findings will be openly accessible upon publication. We have already made the data available in the Supporting Information files and will ensure full access upon acceptance.

Revision made:

• Data Availability Statement rephrased to clarify that data are already available in Supporting Information and will remain accessible upon publication.

7. Copyright issue (Figure 2)

Response:

We carefully reviewed the copyright issue related to Figure 2. To comply fully with PLOS ONE’s licensing requirements, we have removed the copyrighted figure (originally labeled Figure 2-1) from the revised manuscript. No new figure was introduced to replace it. The corresponding text has been updated to maintain clarity.

Revision made:

• Figure 2-1: Removed copyrighted figure; no replacement figure introduced.

• Figure captions have been updated accordingly to reflect the removal of Figure 2-1.

8. Suggested citations

Response:

We reviewed the suggested references and added additional relevant citations where appropriate. These references strengthen the manuscript by supporting statements about IDDSI functional testing, dysphagia nutrition management, and texture–swallow safety relationships.

Revision made:

• Discussion Section: Added several citations in support of texture-modification strategies and the relationship between nutritional intake and safe swallowing. The reference list has been updated accordingly.

Responses to Reviewer #1

We sincerely appreciate Reviewer #1’s positive feedback and insightful suggestions, which have greatly enhanced the clarity and precision of the manuscript.

1. Improve clarity in the Discussion section

Response:

We have revised the Discussion section to improve clarity and avoid unnecessary over-interpretation. The text now explicitly clarifies that the “twin subtype” terminology refers to two different functional pathways within the same IDDSI level, rather than suggesting a redefinition of IDDSI levels. We also reduced speculative language and framed the findings more cautiously as exploratory rather than conclusive.

Revision made:

• Discussion Section: Revised language in 4.1 to clearly define the twin subtypes and avoid over-claiming about IDDSI level equivalence. The tone has been moderated throughout to improve clarity and reduce potential over-interpretation.

2. Add 1-2 sentences in the Conclusion on how nutrition is enhanced for dysphagia

Response:

We have added statements in the Conclusion to explain how the study’s findings on texture–nutrition differentiation may help optimize nutritional intake in dysphagia populations by promoting targeted fortification strategies while maintaining swallowing safety. This provides a clear link between texture characteristics and nutritional support.

Revision made:

• Conclusion Section (5): Added sentences to highlight the potential for pathway-oriented texture management to support improved energy and protein intake.

3. State limitations of the study

Response:

We included a limitations section in the Conclusion acknowledging that the study is based on controlled food models and standardized texture measurements, rather than clinical swallowing validation. We also note that further studies using clinical assessments (e.g., VFSS/FEES) are needed to validate these findings in real-world settings.

Revision made:

• Conclusion Section (5): Added a brief limitations statement regarding the study design and the need for further validation.

4. Clarify VFSS/FEES definition

Response:

We have expanded the first mention of VFSS and FEES to provide their full names and a brief description, as requested. We clarified that these are imaging-based examinations used to assess swallowing physiology and aspiration risk.

Revision made:

• Conclusion Section (5): Added the full definitions of VFSS and FEES and a brief explanation of their purpose in clinical swallowing assessment.

5. Consistent use of “food-ingredients” terminology

Response:

We have consistently used “food ingredients” throughout the manuscript to ensure clarity and consistency, as requested.

Revision made:

• Whole manuscript: Changed "ingredients" to "food ingredients" where appropriate, especially in the Methods and Discussion sections.

6. Address comment bubbles in the manuscript

Response:

We have reviewed and addressed all comments in the manuscript, and the comment bubbles in the tracked-changes version have been resolved.

Revision made:

• Whole manuscript: Comment bubbles have been removed and addressed as appropriate.

7. Improve Table 3-2 formatting (bold IDDSI subtypes)

Response:

We updated Table 3-2 to bold the level comparisons for each of the three pairs of subtypes (MO3 vs LQ3, EX4 vs PU4, EC7 vs RG7), as requested by the reviewer.

Revision made:

• Table 3-2: Updated formatting to highlight the IDDSI subtypes in bold for clarity.

8. Add more relevant citations

Response:

We added relevant citations, particularly in the Discussion section, where additional references support the interpretation of the findings on texture modification, swallowing safety, and nutrition delivery in dysphagia.

Revision made:

• Discussion Section: Added citations to support statements about texture modification strategies and their implications for dysphagia management.

Responses to Reviewer #2

We greatly appreciate Reviewer #2’s thoughtful and thorough critique, which has helped improve the clarity and focus of the manuscript.

1. Clarify IDDSI level equivalence and define “twin subtypes”

Response:

We have clarified that the “twin subtype” terminology does not imply IDDSI reclassification but refers to subtypes within the same IDDSI level. We included a mapping table showing how these subtypes align with official IDDSI functional testing methods and explicitly defined the abbreviations used for the subtypes.

Revision made:

• Methods Section 2.4: Added definitions of MO, LQ, EX, PU, EC, RG and mapping to official IDDSI terminology.

• Discussion Section 4.1: Clarified the distinction between functional pathways and emphasized the role of IDDSI functional tests.

2. Clarify web crawling usage and relation to lab-based texture testing

Response:

We clarified that web crawling was used solely for identifying commonly consumed food ingredients and dietary context. All experimental texture and nutritional measurements were performed on laboratory-prepared samples. This distinction is now clearly stated in the Methods section.

Revision made:

• Methods Section 2.1: Clarified the role of web-derived data for ingredient selection only, not for experimental measurement.

• Methods Section 2.2: Added clarification that all texture measurements were conducted on laboratory-prepared food samples.

3. Address representativeness of selected ingredients

Response:

We emphasized that the selection of ingredients was exploratory and aimed at providing a culturally relevant set of food ingredients for the study, not a statistically representative national sample. This distinction is made to avoid the suggestion of random sampling.

Revision made:

• Methods Section 2.2: Clarified that the ingredient panel was exploratory and culturally representative rather than probabilistic.

4. Provide details on reproducibility and TPA settings

Response:

We have expanded the Methods section to include detailed descriptions of sample preparation standardization, water content control, temperature control, and replicate measurements. TPA testing conditions and sample equilibration time are now explicitly detailed to ensure reproducibility.

Revision made:

• Methods Section 2.3–2.5: Detailed TPA settings, sample prep standardization, and water content control.

5. Nutrition estimation validity and dilution effects

Response:

We addressed concerns regarding the nutritional estimation method by explaining that the values were derived from food composition databases and standardized cooking procedures. We also clarified that hydration effects are controlled experimentally, and we introduced additional density indicators to account for nutrient delivery, avoiding over-reliance on per-100g comparisons.

Revision made:

• Methods Section 2.6: Added clarification on nutrient density estimation and hydration effects.

• Discussion Section 4.2: Introduced new density-related indicators (e.g., kcal/g and protein density) to improve nutrient interpretation.

6. Reframe conclusions and propose future validation

Response:

We revised the conclusion to emphasize that the study’s findings suggest potential clinical implications rather than definitive clinical guidance. We added a statement proposing future validation studies, including VFSS/FEES and clinical outcomes, to support broader clinical applications.

Revision made:

• Conclusion Section 5: Reframed conclusions and added a statement about future clinical validation.

We appreciate the detailed feedback provided by the reviewers and the editor. We believe these revisions have strengthened the manuscript and improved its clarity, relevance, and scientific rigor. We hope the revised version will be suitable for publication in PLOS ONE.

Kind regards,

Dr. Muxi Chen

(on behalf of all authors)

---

## [Decision Letter · Decision Letter 1]

22 Feb 2026

Objective Differences and Pathway Differentiation between Twin Subtypes within the Same IDDSI Level: A Texture–Nutrition Analysis Based on Commonly Consumed Foods among Chinese Elderly

PONE-D-25-62691R1

Dear Dr. Chen,

We’re pleased to inform you that your manuscript has been judged scientifically suitable for publication and will be formally accepted for publication once it meets all outstanding technical requirements.

Kind regards,

António Raposo

Academic Editor

PLOS One

Additional Editor Comments (optional):

Reviewers' comments:

Reviewer's Responses to Questions

**Comments to the Author**

Reviewer #1: All comments have been addressed

Reviewer #2: All comments have been addressed

2. Is the manuscript technically sound, and do the data support the conclusions?

Reviewer #1: Yes

Reviewer #2: Yes

3. Has the statistical analysis been performed appropriately and rigorously?

Reviewer #1: Yes

Reviewer #2: Yes

4. Have the authors made all data underlying the findings in their manuscript fully available?

Reviewer #1: Yes

Reviewer #2: Yes

5. Is the manuscript presented in an intelligible fashion and written in standard English?

Reviewer #1: Yes

Reviewer #2: Yes

Reviewer #1: I suggest that the last paragraph should be re-phrased as:

"Overall, this study contributes an objective evidence for integrating instrumental texture metrics with

nutritional considerations within the IDDSI framework. In addition, it supports a more nuanced,

pathway-oriented approach to dysphagia nutrition care".

Reviewer #2: After seeing the improvements made in the text, namely in the discussion, I agree that this version can now be published.

.

Reviewer #1: **Yes:** Dele RaheemDele RaheemDele RaheemDele Raheem

Reviewer #2: **Yes:** M. João LimaM. João LimaM. João LimaM. João Lima

---

## [Editor Report · Acceptance letter]

PONE-D-25-62691R1

PLOS One

Dear Dr. Chen,

I'm pleased to inform you that your manuscript has been deemed suitable for publication in PLOS One. Congratulations! Your manuscript is now being handed over to our production team.

Kind regards,

on behalf of

Dr. António Raposo

Academic Editor

PLOS One